# Fair Classifiers that Abstain without Harm

**Tongxin Yin[1]\*, Jean-François Ton[2], Ruocheng Guo[2], Yuanshun Yao[2], Mingyan Liu[1], Yang Liu[2]**
[1] University of Michigan
[2] ByteDance Research
tyin@umich.edu, jeanfrancois@bytedance.com, rguo.asu@gmail.com,
kevin.yao@bytedance.com, mingyan@umich.edu, yang.liu01@bytedance.com

## Abstract

In critical applications, it is vital for classifiers to defer decision-making to humans. We propose a post-hoc method that makes existing classifiers selectively abstain from predicting certain samples. Our abstaining classifier is incentivized to maintain the original accuracy for each sub-population (i.e. no harm) while achieving a set of group fairness definitions to a user specified degree. To this end, we design an Integer Programming (IP) procedure that assigns abstention decisions for each training sample to satisfy a set of constraints. To generalize the abstaining decisions to test samples, we then train a surrogate model to learn the abstaining decisions based on the IP solutions in an end-to-end manner. We analyze the feasibility of the IP procedure to determine the possible abstention rate for different levels of unfairness tolerance and accuracy constraint for achieving no harm. To the best of our knowledge, this work is the first to identify the theoretical relationships between the constraint parameters and the required abstention rate. Our theoretical results are important since a high abstention rate is often infeasible in practice due to a lack of human resources. Our framework outperforms existing methods in terms of fairness disparity without sacrificing accuracy at similar abstention rates.

## 1 Introduction

Enabling machine learning (ML) systems to abstain from decision-making is essential in high-stakes scenarios. The development of classifiers with appropriate abstention mechanisms has recently attracted significant research attention and found various applications (Herbei & Wegkamp, 2006; Cortes et al., 2016; Madras et al., 2018a; Lee et al., 2021; Mozannar et al., 2023). In this paper, we demonstrate that allowing a classifier to abstain judiciously enhances fairness guarantees in model outputs while maintaining, or even improving, the accuracy for each sub-group in the data.

Our work is primarily anchored in addressing the persistent dilemma of the fairness-accuracy tradeoff – a prevalent constraint suggesting that the incorporation of fairness into an optimization problem invariably compromises achievable accuracy (Kleinberg et al., 2016). To circumvent this problem, we propose to use classifiers with abstentions. Conventionally, the fairness-accuracy tradeoff arises due to the invariance of *data distribution* and the rigid *model hypothesis space*. Intuitively, by facilitating abstentions within our model, we introduce a framework that permits the relaxation of both limiting factors (distribution & model space). This transformation occurs as the abstention mechanism inherently changes the distributions upon which the classifier's accuracy and fairness are computed. In addition, since the final model output is a combination of the abstention decision and the original model prediction, the model hypothesis space expands. This adaptability paves the way for our approach to breaking the fairness-accuracy curse.

There exist several works that explore abstaining classifiers to achieve better fairness (Madras et al., 2018b; Lee et al., 2021). In contrast, we aim to achieve the following four requirements simultaneously, a rather ambitious goal that separates our work from the prior ones:

- **Feasibility of Abstention**: We need to determine if achieving fairness with no harm is feasible or not at a given abstention rate.

---

\*Part of the work is done as an intern at Bytedance.

| Related Works | Abstention Rate Control | Multiple Fairness | Fairness Guarantee | No Harm |
|---|---|---|---|---|
| LTD (Madras et al., 2018b) | | ✓ | | |
| FSCS (Lee et al., 2021) | ✓ | | | |
| FAN (Our work) | ✓ | ✓ | ✓ | ✓ |

Table 1: A summary of key properties of our work and closely related works.

- **Compatible with Multiple Common Fairness Definitions**: We seek a flexible solution that can adapt to different fairness definitions.
- **Fairness Guarantee**: We aim for a solution that provides a strong guarantee for fairness violations, i.e., impose hard constraint on disparity.
- **No Harm**: We desire a solution that provably guarantees each group's accuracy is no worse than the original (i.e. abstaining-free) classifier.

We propose a post-hoc solution that abstains from a given classifier to achieve the above requirements. Our solution has two stages. In Stage I, we use an integer programming (IP) procedure that decides whether it is feasible to satisfy all our requirements with a specific abstention rate. If feasible, Stage I will return the optimal abstention decisions for each training sample. However, a solution that satisfies all our constraints might not exist. To expand the feasible space of the solution, we also selectively flip the model prediction. Stage I informs us how to abstain on the training samples; to expand the abstention (and flipping) decisions to unseen data, Stage II trains a surrogate model to encode and generalize the optimal abstention and flipping patterns in an end-to-end manner.

We name our solution as **F**air **A**bstention classifier with **N**o harm (`FAN`). Compared to the prior works, our solution guarantees the four desired properties mentioned before, shown in Table 1. To the best of our knowledge, our method is the first to develop an abstention framework that incorporates a variety of constraints, including *feasible abstention rates*, *compatibility with different fairness definition*, *fairness guarantees* and *no harm*. We theoretically analyze the conditions under which the problem is feasible - our work is the first to characterize the feasibility region for an abstaining mechanism to achieve some of the above-listed constraints. We have carried out extensive experiments to demonstrate the benefits of our solution compared to strong existing baselines.

## 1.1 RELATED WORK

**Fair Classification.** Our work relates broadly to fairness in machine learning literature (Dwork et al., 2012; Hardt et al., 2016; Kusner et al., 2017; Menon & Williamson, 2018; Ustun et al., 2019). Our work is particularly inspired by the reported fairness-utility tradeoff (Menon & Williamson, 2018; Kleinberg et al., 2016). One way to resolve the problem is to decouple the training of classifiers to guarantee each group receives a model that is no worse than the baseline but this line of work often requires knowing the sensitive attribute at test time and is less flexible to incorporate different fairness definitions (Ustun et al., 2019). There's a wide range of approaches available to achieve fairness, including pre-processing methods (Nabi & Shpitser, 2018; Madras et al., 2018a; Luong et al., 2011; Kamiran & Calders, 2012), in-processing techniques (Pleiss et al., 2017; Noriega-Campero et al., 2019; Kim et al., 2018), and post-processing methods (Ustun et al., 2019; Agarwal et al., 2018; Kamishima et al., 2012). Our work specifically focuses on post-processing techniques.

**Abstain Classifier.** Existing literature provides an expansive exploration of abstention or selective classifiers (Cortes et al., 2016; Chow, 1957; Hellman, 1970; Herbei & Wegkamp, 2006; Geifman & El-Yaniv, 2017). Typically, selective classification predicts outcomes for high-certainty samples and abstains on lower ones, where the softmax outputs of the classifier are employed (Cordella et al., 1995; El-Yaniv et al., 2010). Interestingly, (Jones et al., 2020) highlights a potential pitfall, suggesting that selective classifiers can inadvertently exacerbate fairness problems if not used judiciously. This finding underscores the importance of careful application and has inspired various fair selective methodologies (Shah et al., 2022; Lee et al., 2021; Schreuder & Chzhen, 2021; Mozannar & Sontag, 2020; Madras et al., 2018b). However, these methodologies primarily focus on regression or incorporate fairness constraints in their optimization objectives to create selective classifiers. For instance, `LTD` (Madras et al., 2018b) introduces a penalty term to address high abstention rates and unfairness. However, it lacks robust mechanisms for controlling both abstention rates and fairness. On the other hand, `FSCS` (Lee et al., 2021) presents an abstention framework specifically designed to reduce precision disparities among different groups, but it does not accommodate other fairness

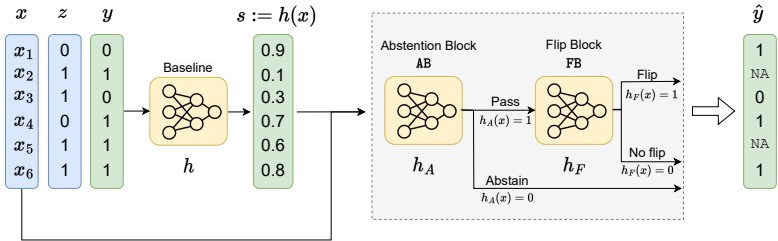

Figure 1: Overview of FAN. We first get the baseline classifier $h$'s confidence score on data, i.e. $s = h(x)$. We then forward the confidence scores to Abstention Block AB where a model $h_A$ either tells us to abstain (i.e. $h_A(x, s) = 0$) or to pass to the Flip Block FB (i.e. $h_A(x, s) = 1$). If abstained, then it is the final outcome. Otherwise, FB will decided on the unabstained samples if their predicted labels $\hat{y}_b = s \geq t_0$ should be flipped or not. $\hat{y}$ is the final outcome.

definitions. Additionally, neither of these approaches offers a way to monitor or control the accuracy reduction for each group.

Our paper proposes a novel approach that first utilizes exact integer programming to establish the optimal classifier that satisfies all the aforementioned constraints, then trains a surrogate model on the output of said IP. The most relevant work to ours is (Mozannar et al., 2023), which applies Mixed-Integer Programming (MIP) to the selective classification problem, but differs in terms of the fairness, no harm, and feasibility guarantees. Furthermore, we introduce distinctive strategies that can be deployed without requiring knowledge of the true label. These strategies involve training a model based on the IP's output, which not only enhances efficiency but also substantially reduces computational requirements, especially in large-scale problems. Whereas the MIP design proposed in Mozannar et al. (2023) is limited to moderately-sized problems and relies on the availability of true labels at the inference time.

## 2  PRELIMINARIES AND OVERVIEW

Let $\mathcal{D}$ be a data distribution defined for a set of random variables $(X, Z, Y)$, representing each feature (e.g., application profile in a loan application), protected attribute (e.g., gender or race), and label (qualified or not in a loan application), respectively. Consider a discrete $Z \in \mathcal{Z}$ and a binary classification problem where $Y = 1$ indicates the model predicts positive (i.e. to the favorable outcome) on the sample, $Y = 0$ indicates negative (i.e., unfavorable), and $X \in \mathcal{X}$. We assume a training dataset sampled i.i.d. from $\mathcal{D}$ with $N$ samples: $(x_1, z_1, y_1), (x_2, z_2, y_2), \cdots, (x_N, z_N, y_N)$. We aim to develop a post-hoc framework, named FAN, that takes in a trained classifier $h : \mathcal{X} \to [0, 1]$, i.e. the baseline classifier, and outputs its abstaining decisions. Denote $S = h(X) \in [0, 1]$ the *confidence score* for individual $X$, and the predicted label $\hat{Y}_b = 1[h(X) \geq t_0]$ based on a user-specified threshold $t_0$.

Figure 1 shows the overview of FAN. We will use two modules, Abstention Block (AB) and Flip Block (FB), to determine from which samples the classifier should abstain. The goal of AB is to decide which samples to abstain in order to satisfy our set of constraints; the goal of FB is to expand the feasibility region of the final decision outcomes, enabling a larger feasibility region of the formulated problem (see Section 3.1 for the explanation).

AB, i.e. $h_A : [\mathcal{X}, h(\mathcal{X})] \to \{0, 1\}$, takes the feature $X$ of the given individual, and the corresponding confidence score $S$ predicted from the baseline model as inputs, and decides whether to abstain the prediction. $h_A(X, h(X)) = 0$ indicates that the prediction should abstain. Samples that are not abstained by AB will be forwarded to FB. FB, i.e. $h_F : [\mathcal{X}, h(\mathcal{X})] \to \{0, 1\}$ decides whether to flip the prediction of $h$ or not, which is the final decision of FAN:

$$\hat{Y} = \begin{cases} 1 - \hat{Y}_b & \text{if } h_F(X, h(X)) = 1 \\ \hat{Y}_b & \text{otherwise} \end{cases} \tag{1}$$

## 3 METHOD

We explain how we formulate the problem to achieve our goal and our two-stage algorithm.

### 3.1 PROBLEM FORMULATION

In general, improving fairness often results in decreased accuracy (Kleinberg et al., 2016). In our case, though, we enable abstention, which allows us to prioritize fairness while still maintaining accuracy. Furthermore, we desire a formulation that imposes hard constraints for both fairness and accuracy, as compared to prior works that only incorporate a soft penalty term into the objective function (Madras et al., 2018b; Lee et al., 2021).

Specifically, we use the following optimization problem to obtain $h_A, h_F$ in our AB and FB:

$$\min_{h_A, h_F} \quad \mathbb{E}_{\mathcal{D}}\left[\left(h_F(X, h(X))(1 - \hat{Y}_b) + (1 - h_F(X, h(X)))\hat{Y}_b\right) \neq Y \mid h_A(X, h(X)) = 1\right] \quad \textbf{(Error Rate)}$$

$$\text{s.t.} \quad \mathscr{D}(h_A, h_F, z, z') \leq \mathscr{E}, \forall z, z' \in \mathcal{Z} \quad \textbf{(Disparity)}$$

$$\mathbb{E}_{\mathcal{D}}\left[h_A(X, h(X)) \mid Z = z\right] \geq 1 - \delta_z, \forall z \in \mathcal{Z} \quad \textbf{(Abstention Rate)}$$

$$\mathbb{E}_{\mathcal{D}}\Big[\left(h_F(X, h(X))(1 - \hat{Y}_b) \right.$$

$$\left. + (1 - h_F(X, h(X)))\hat{Y}_b\right) \neq Y \mid h_A(X, h(X)) = 1, Z = z\Big] \leq e'_z, \forall z \in \mathcal{Z}, \quad \textbf{(No Harm)}$$

where $e'_z = (1 + \eta_z)e_z$. $e_z = \mathbb{E}_{\mathcal{D}}[h(X) \neq Y \mid Z = z]$ is the error rate of baseline optimal classifier $h$, $\delta_z$. $\eta_z$ is a "slack" we allow for the no harm constraint and is chosen such that $0 \leq (1 + \eta_z)e_z \leq 1$.

**Error Rate.** Our main objective is to minimize 0-1 loss for all samples that are not abstained.

**Disparity.** We enforce a fairness constraint between every pair of groups $z, z' \in \mathcal{Z}$, by bounding the disparity $\mathscr{D}$ within the predefined design parameter $\mathscr{E}$. There are several fairness definitions that can be applied. In this paper, we utilize three specific fairness notions, Demographic Parity (DP) (Dwork et al., 2012), Equal Opportunity (EOp) (Hardt et al., 2016), Equalized Odds (EOd) (Hardt et al., 2016). Details are shown in Table 3.

**Abstention Rate.** Although abstention can lead to better model performance, a high abstention rate can be impractical due to a lack of human resources. Therefore, it is crucial to limit the abstention rate. To address this issue, we set a maximum threshold for the proportion of instances that the system can abstain in each group. The abstention rate should not exceed a user-specified threshold $\delta_z$ for each group $z$. Intuitively, this means that we cannot simply decide to forgo giving predictions on the majority of the samples (or the majority from a certain group), because even though it would satisfy all the other constraints it would not be practically useful. Note that to introduce more flexibility, we enable independent control of the abstention rates for each group.

**No Harm.** We ensure the classifier does not compromise the accuracy of the groups. The extent of relaxation is determined by a user-specified $\eta_z$, which establishes the maximum allowable reduction in accuracy. When $\eta_z > 0$, IP permits a certain degree of relaxation on the error rate bound for each group. Conversely, when $\eta_z < 0$, it implies that a lower group error rate is mandated.

**Why Need FB.** The fairness and no harm constraints specified in **Disparity** and **No Harm** jointly impose challenging constraints for the decision rule to satisfy. For instance, the no harm constraint only allows certain predictions to be abstained, as this constraint essentially requires us to abstain more from wrongly predicted samples. When a classifier is relatively accurate, and when the abstention rate is constrained, we are left with only a small feasibility region. The FB block opens up more design space for the abstention policy, as we flip properly, the disparity and no harm conditions could become easier to satisfy. Note that flipping model predictions is a popular post hoc way of expanding the model decision space towards improving fairness (Menon & Williamson, 2018). We illustrate it using the following example:

**Example 3.1.** Consider Demographic Parity (DP) as the fairness measure, imagine a system with two groups, where the allowed abstention rate is $\delta_1 = \delta_2 = 0.1$. If we set $\varepsilon = 0.1$ as the permissible disparity in demographic parity (DP), according to the baseline classifier, the acceptance rate for group 1 and 2 are 0.3 and 0.7 respectively. Even if we abstain only from the positive samples in group 2, the adjusted acceptance rates would be 0.3 and 0.6 respectively, while the resulting disparity

(0.3) is still greater than $\varepsilon$. However, if flipping is allowed, we can further flip 0.2 positive samples of group 2 to negative, resulting in final adjusted acceptance rates of 0.3 and 0.4. *

## 3.2 Two-Stage Procedure

Directly solving the optimization problem in Section 3.1 is challenging because it would require joint training of $h_A$ and $h_F$. In addition, the analysis of its feasibility would also highly rely on the hypothesis space for learning $h_A$ and $h_F$. Lastly, the composition of multiple sets of constraints adds to the difficulty of solving and analyzing it. To solve those challenges, we propose a *two-stage approach* to train $h_A$ and $h_F$. Instead of solving the inflexible and costly optimization problem on the fly, it learns the optimal abstention patterns end to end.

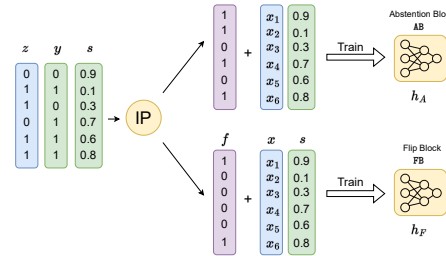

Figure 2: Illustration of the two-stage design.

**Stage I: Integer Programming**. For a dataset with $N$ individuals, the goal of Stage I is to learn two N-length vectors: $\omega = \{\omega_n\}_N$ and $f = \{f_n\}_N$, which represent predictions of $h_A(X, h(X))$ and $h_F(X, h(X))$ on this dataset, respectively. In other words, $\omega_n = h_A(x_n, h(x_n)) \in \{0,1\}$, and $f_n = h_F(x_n, h(x_n)) \in \{0,1\}$.

$$\min_{\omega, f} \quad \sum_{n=1}^{N} \omega_n \cdot 1[\hat{y}_n \neq y_n] := \sum_{n=1}^{N} \omega_n \cdot 1\left[(\hat{y}_{bn}(1 - f_n) + (1 - \hat{y}_{bn})f_n) \neq y_n\right] \qquad \textbf{(IP-Main)}$$

$$\text{s.t.} \quad \bar{\mathcal{D}} \leq \bar{\mathcal{E}}, \forall z, z' \in \mathcal{Z} \qquad \textbf{(Disparity)}$$

$$\frac{\sum_{n=1}^{N} \omega_n \cdot 1[z_n = z]}{\sum_{n=1}^{N} 1[z_n = z]} \geq (1 - \delta_z), \forall z \in \mathcal{Z} \qquad \textbf{(Abstention Rate)}$$

$$\sum_{n=1}^{N} \omega_n \cdot 1[\hat{y}_n \neq y_n, z_n = z] \leq \left(\sum_{n=1}^{N} \omega_n \cdot 1[z_n = z]\right) \cdot (1 + \eta_z)e_z, \forall z \in \mathcal{Z} \qquad \textbf{(No Harm)}$$

$$\omega_n \in \{0,1\}, f_n \in \{0,1\}, \forall n.$$

Solving it gives us the abstention (i.e. $\omega$) and flipping decision (i.e. $f$) for each of the training data. The empirical version of the (**Disparity**) constraints can be found in Table 4 in the Appendix.

**Stage II: Learning to Abstain.** Although IP results offer an optimal solution for the training data, they are not applicable at inference time. This is due to two main reasons. First, accessing the ground truth label $y$ is impossible during inference, which is a necessary input for IP. Second, solving IP is too time-consuming to perform during inference. To solve this problem, we train surrogate models to learn the abstaining and flipping patterns in an end-to-end manner (i.e. from features to abstention and flipping decisions). We use the IP solutions (on the training samples) as the surrogate models' training data, and we want the surrogate model to generalize the patterns to the unseen test samples. Figure 2 illustrates our design and we will describe the details in Appendix B.

Note that we only need to solve **IP-Main** and train surrogate models during the training process, and when we deploy FAN, we only need to run inference on the trained AB and FB, and therefore the inference overhead is small.

## 4 Theoretical Analisis: Feasibility and Fairness in Abstention

The selection of design parameters (including $\delta_z, \eta_z, \varepsilon$), plays a crucial role in training AB and FB, therefore the overall performance of FAN. A higher level of disparity restriction can intuitively result in a higher rate of data samples being abstained from classification, while a more stringent accuracy requirement can also increase the abstention rate and make the problem infeasible. In this section, we focus on theoretically analysis of Stage I, Specifically we answer the following research questions:

---

*To keep the example simple, we do not consider accuracy here. While in our formulation, **Error Rate** and **No Harm** together ensure that flipping would not cause harm but rather incentivize improvements in accuracy.

**Under what conditions will Problem IP-Main become feasible for each fairness notion? What is the relationship between the design parameters?**

We derive the feasibility condition for the IP formulation (**IP-Main**) in Stage I. The task of theoretically bounding the performance gap between predictions (surrogate models in Stage II) and ground truth (IP solution in Stage I) is generally challenging as the models are neural networks, therefore we study it empirically in Section 5.

We summarize the key parameters used in this section:

| $\varepsilon$ | $\delta_z$ | $e_z$ | $\eta_z$ | $\tau_z$ (TBD in 4.1) |
|---|---|---|---|---|
| Fairness Disparity | Abstention rate allowed for $z$ | Error rate for $z$ by $h$ | Error rate slack or restrictiveness compared to baseline | Qualification rate of group $z$ |

### 4.1 FEASIBILITY

Define $\tau_z = \frac{\sum_n 1[z_n = z] y_n}{\sum_n 1[z_n = z]}$ the proportion of qualified individuals of group $z$, i.e., qualification rate of group $z$. We prove the following results for demographic parity:

**Theorem 4.1.** *(Feasibility of Demographic Parity (DP))* (**IP-Main**) *is feasible under DP if and only if* $\forall \bar{z}, \underline{z} \in \mathcal{Z}$ *such that* $\tau_{\bar{z}} \geq \tau_{\underline{z}}$,

$$\delta_{\bar{z}} \geq 1 - \frac{1 + \varepsilon + (1 + \eta_{\underline{z}})e_{\underline{z}} - \tau_{\bar{z}} + \tau_{\underline{z}}}{1 - (1 + \eta_{\bar{z}})e_{\bar{z}}}. \tag{2}$$

Theorem 4.1 demonstrates the feasibility of the IP to achieve Demographic Parity. Specifically, the theorem establishes the minimum value of $\delta_z$ that is allowed, subject to upper bounds on disparity and a relaxation parameter for the error rate. This highlights the importance of abstention by the more qualified group (higher qualification rate) for achieving a fair model without compromising accuracy, while the less qualified group need not abstain. Later in Section 4.2, we provide further treatment to remedy the concern over an imbalanced abstention rate.

Specifically, for the two group scenario ($\mathcal{Z} = \{\bar{z}, \underline{z}\}$), our results demonstrate that increasing the values of $\eta_{\bar{z}}$ and $\eta_{\underline{z}}$ will lead to smaller values of $\delta_{\bar{z}}$, indicating that a relaxation of the error rate can allow the more qualified group to abstain from fewer samples. Additionally, a looser bound on disparity will also enable the more qualified group to abstain from fewer samples. In practice, determining an appropriate value of $\delta_{\bar{z}}$ is of paramount importance. To this end, we present the following illustrative example.

**Example 4.2.** a) If $\tau_{\bar{z}} = \tau_{\underline{z}}$, i.e., the dataset is balanced, and $(1 + \eta_{\bar{z}})e_{\bar{z}} < 1$, we have that $1 - \frac{1 + \varepsilon + (1 + \eta_{\underline{z}})e_{\underline{z}} - \tau_{\bar{z}} + \tau_{\underline{z}}}{1 - (1 + \eta_{\bar{z}})e_{\bar{z}}} < 0$, therefore the problem is always feasible. b) If $\tau_{\bar{z}} - \tau_{\underline{z}} = 0.3$, $e_{\bar{z}} = e_{\underline{z}} = 0.1, \eta_{\bar{z}} = \eta_{\underline{z}} = 0$, when $\varepsilon = 0.05, \delta_{\bar{z}} \geqslant 0.056$; when $\varepsilon = 0.1, \delta_{\bar{z}}$ has no restriction.

Further for Equal Opportunity and Equalized Odds we have the following results:

**Theorem 4.3.** *(Feasibility of Equal Opportunity (EOp))* **IP-Main** *is always feasible under EOp.*

**Theorem 4.4.** *(Feasibility of Equalized Odds (EOd))* **IP-Main** *is always feasible under EOd.*

Theorems 4.3 and 4.4 demonstrate the feasibility of the IP under Equal Opportunity and Equalized Odds. Specifically, regardless of the design parameters' values, our results indicate that a feasible solution to the IP problem always exists. Notably, our results imply that even when the abstention rate is 0, the IP can solely adjust the flip decisions $f_n$ to satisfy constraints on disparate impact, abstention rate, and no harm. More discussion on this can be found in Appendix C.

### 4.2 EQUAL ABSTENTION RATE

An objection may arise that the model's excessive abstention from a particular group, while not abstaining from others. Moreover, if such abstention occurs solely on data samples with positive or negative labels, further concerns may be raised. In this section, we delve into a scenario where differences in abstention rates across groups and labels are constrained. We show that under equal abstention rate constraints, the performance of IP will become worse (higher overall error rate)

compared to Problem **IP-Main**.

$$\min_{\omega,f} \quad \textbf{IP-Main} \tag{3}$$

$$\text{s.t.a.} \quad \left| \frac{\sum_{n=1}^N \omega_n 1[z_n = z, y_n = y]}{\sum_{n=1}^N 1[z_n = z, y_n = y]} - \frac{\sum_{n=1}^N \omega_n 1[z_n = z', y_n = y]}{\sum_{n=1}^N 1[z_n = z', y_n = y]} \right| \leq \sigma_y, \forall z, z' \in \mathcal{Z}, y \in \{0,1\}$$

**Theorem 4.5.** *(Feasibility of Demographic Parity with Constraint Disparity of Abstention Rate) A sufficient condition for Problem 3 being feasible is $\forall \bar{z}, \underline{z} \in \mathcal{Z}$ such that $\tau_{\bar{z}} \geq \tau_{\underline{z}}$,*

$$\delta_{\bar{z}} \leq \tau_{\bar{z}}\sigma_1, \quad \delta_{\underline{z}} \leq \tau_{\underline{z}}\sigma_1, \quad \delta_{\bar{z}} \geq 1 - \frac{1 + \varepsilon + (1 + \eta_{\underline{z}})e_{\underline{z}} - \tau_{\bar{z}} + \tau_{\underline{z}}}{1 - (1 + \eta_{\bar{z}})e_{\bar{z}}}. \tag{4}$$

We similarly show that for Equal Opportunity and Equalized Odds the problem remains feasible even under equal abstention rate constraints. We defer these details to Appendix.

## 5 EXPERIMENTS

In this section, we evaluate `FAN` using various real-world datasets, comparing it against baselines and to better understand its components. We start by explaining our experimental settings and then move on to how `FAN` performs in comparison to other methods. We also analyze the separate components of `FAN` to get a clearer picture of how each contributes to the overall performance. Additionally, we compare our trained models, specifically `AB` and `FB`, with integer programming (IP) solutions. [†]

In our study, we primarily focus on a setting involving only two distinct groups. We set the same abstention rate across all groups, i.e., $\delta_z = \delta$. `FAN` is evaluated against two baselines: `LTD` (Madras et al., 2018b) and `FSCS` (Lee et al., 2021), shown in Table 1. For `LTD`, we employ the learning-to-reject framework, specifically referring to Equation 4 in (Madras et al., 2018b)[‡]. We adopt three real-world datasets: `Adult` (Dua & Graff, 2017), `Compas` (Bellamy et al., 2018), and `Law` (Bellamy et al., 2018). During training, Equalized Odds incorporate two separate constraints, while to facilitate a straightforward interpretation, the average disparity is shown. The details of data preprocessing and model setting can be found in Appendix E.

**Baseline Optimal.** For `FAN`, we use an optimal classifier trained solely to minimize the loss as the baseline $h$, naming it "baseline optimal". We use Multi-Layer Perceptron (MLP) to train baseline optimal, `AB`, and `FB`. Details can be found in Appendix E. Table 6 in Appendix shows the performance of the baseline optimal model on both the training and test datasets (including overall accuracy and group accuracy, along with disparities measured under DP, EOp, and EOd.) Specifically, the overall training accuracy on `Adult`, `Compas` and `Law` are 92.08%, 72.33%, 82.86%, respectively.

**Overall Performance.** Figure 3 illustrates how `FAN` compares to `LTD` and `FSCS` across different datasets and abstention rates. In the `LTD` method, abstention is introduced as a penalty term in the objective function, making it difficult to precisely control the abstention rate. To work around this, we adjust the penalty term's coefficient and chart the resulting actual abstention rate. The first row of the figure highlights the disparity reduction each algorithm achieves compared to the baseline optimal $h$. The second row shows the *minimum increase in group accuracy* for all groups. Generally, `FAN` yields the most significant reduction in disparity without sacrificing much accuracy, unlike `FSCS` and `LTD`, which focus more on fairness at the cost of accuracy. Using the no-harm constraint **No Harm**, `FAN` often matches or even surpasses the baseline optimal classifier in terms of accuracy. Nevertheless, there are a few instances where accuracy slightly drops, which we discuss further below.

**Stage II Analysis: Performance of Surrogate Model.** The no-harm constraint is imposed to encourage `FAN` to maintain or even improve group-level accuracy when compared to the baseline. The integer programming formulation in Equation **IP-Main** is designed to strictly satisfy this constraint. However, `FAN` may not strictly meet this due to the surrogate model training of `AB` and `FB` in what we refer to as Stage II. As seen in the second row of Figure 3, there are instances where accuracy slightly decreases. Table 2 provides insights into this by illustrating the training accuracy of `AB` and `FB`. This suggests that the surrogate models are effective at learning from the IP outcomes. Figure 17 also shows the loss of `AB` and `FB` on `Adult` under Demographic parity, as an example.

---

[†]Code: `https://github.com/tsy19/FAN`

[‡]Learning-to-defer schema requires an additional Decision Maker, which does not apply to our focal scenario.

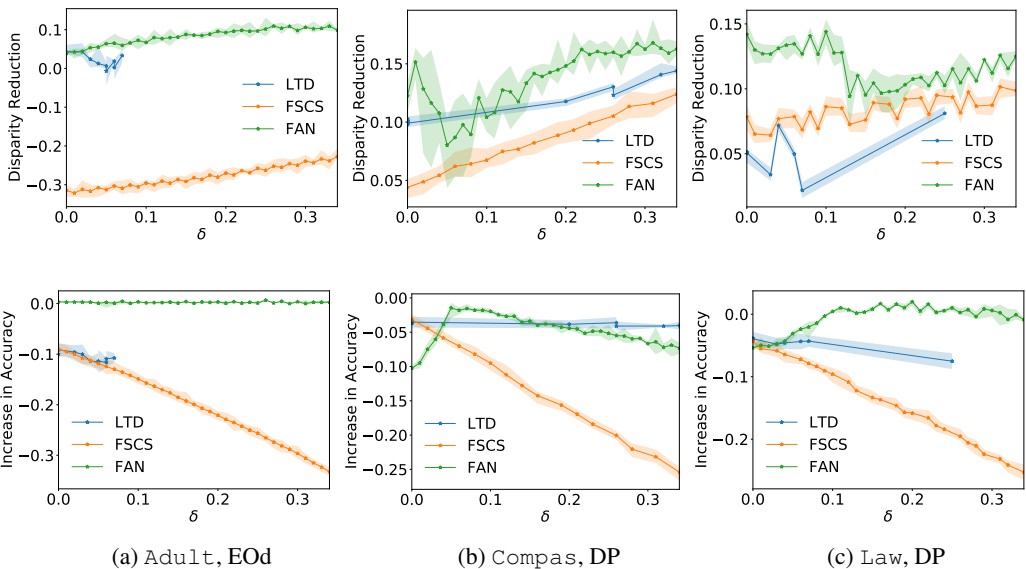

(a) Adult, EOd      (b) Compas, DP      (c) Law, DP

Figure 3: Comparison of FAN with baseline algorithms. The 1st row depicts disparity reduction, and the 2nd row presents the minimum group accuracy increase, both compared to baseline optimal. (a) Evaluation on the Adult under Equalized Odds. (b) Evaluation on the Compas under Demographic Parity. (c) Evaluation on the Law under Demographic Parity. For FAN, $\eta_z$ is set to 0.

| Accuracy (%) | | Adult | | | Compas | | | Law | | |
| --- | --- | --- | --- | --- | --- | --- | --- | --- | --- | --- |
| | | DP | EOp | EOd | DP | EOp | EOd | DP | EOp | EOd |
| $\delta = 0.1$ | AB | 94.23 | 93.29 | 93.55 | 90.26 | 90.32 | 95.22 | 96.09 | 91.42 | 92.03 |
| | FB | 94.26 | 94.01 | 94.54 | 79.87 | 79.57 | 76.15 | 88.12 | 91.38 | 90.14 |
| $\delta = 0.2$ | AB | 92.20 | 89.93 | 88.48 | 82.94 | 90.32 | 91.44 | 96.12 | 95.75 | 92.11 |
| | FB | 97.79 | 95.33 | 95.86 | 86.07 | 79.63 | 77.03 | 87.90 | 87.90 | 90.29 |
| $\delta = 0.3$ | AB | 89.94 | 87.42 | 87.43 | 80.28 | 79.99 | 82.82 | 86.50 | 86.39 | 94.82 |
| | FB | 97.18 | 96.31 | 96.33 | 87.72 | 88.55 | 85.53 | 93.00 | 93.92 | 88.17 |

Table 2: Performance Evaluation of Surrogate Model Training. We use MLP as the network structure for both AB and FB. Each cell displays the average training accuracy (from 5 individual runs) of AB (first row) and FB (second row) for specific $\delta$, fairness notion, and dataset employed. Generally, both AB and FB demonstrate a strong ability to learn the IP outcomes effectively and achieve high accuracy, underscoring the success of Stage II. It is worth mentioning that, under some settings, the accuracy on the Compas is low, for example the training accuracy of FB under EOd on Compas with abstention rate 0.1 is 76.15%. However, the issue lies not with our Stage II design but rather with the limitations of the MLP. As demonstrated in Table 6, the performance of the baseline optimal classifier (training accuracy 72.33%) on the Compas is also low.

**Comparison to Baseline Optimal Classifier.** We conduct a comprehensive set of experiments to examine the impact of FAN on both disparity and accuracy. Figure 4 depicts the performance metrics when applying EOp on the training and test data for the Adult dataset. These results offer a comparative benchmark against the baseline optimal classifier $h$, assessing how FAN enhances model performance regarding fairness and accuracy. The figure shows FAN successfully optimizes for a more equitable model without sacrificing accuracy. Notably, as the permissible abstention rate ($\delta$) increases, both groups experience a significant improvement in accuracy, while simultaneously reducing the overall disparity. These findings indicate that FAN has the ability to train models that are both fairer and more effective, particularly when higher levels of abstention are allowed.

**IP Solution.** Figure 5 is the IP solution corresponding to Figure 4. All constraints are strictly satisfied. As $\varepsilon$ increases, the impact on reducing disparity diminishes due to the inherent allowance for greater disparity in the IP formulation. As $\delta$ increases, the accuracy improvement for both demographic

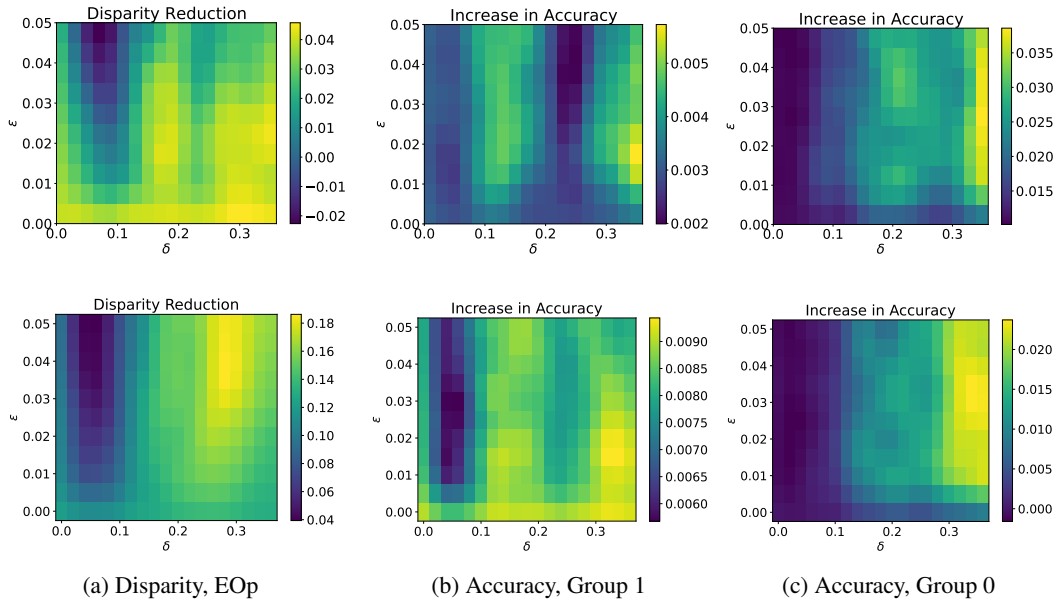

(a) Disparity, EOp  (b) Accuracy, Group 1  (c) Accuracy, Group 0

Figure 4: Disparity reduction and increased accuracy for each group performed on `Adult`, compared to baseline optimal. The first row shows the performance on the training data while the second row is on test data. (a) demonstrates the disparity reduction in terms of EOp, while (b) and (c) showcase the increases in accuracy for both groups. x-axis represents $\delta$, while y-axis represents $\varepsilon$.

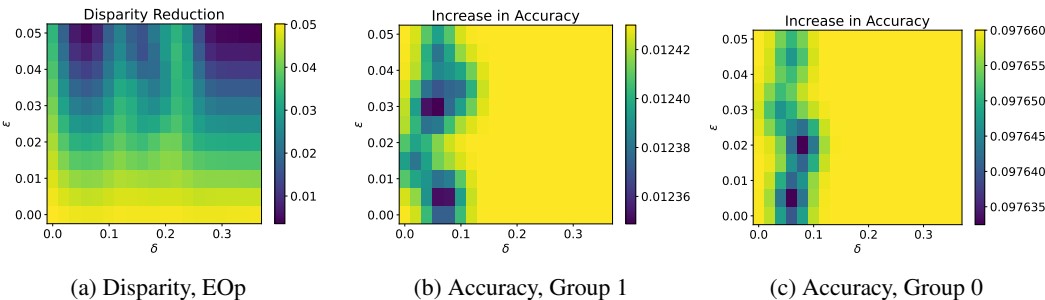

(a) Disparity, EOp  (b) Accuracy, Group 1  (c) Accuracy, Group 0

Figure 5: This figure illustrates the result of the IP solution, under the same setting of Figure 4.

groups stays stable across configurations, mainly due to already nearing optimality in this experiment. Comparison between Figure 5 and Figure 4 underscores the successful learning of the IP solution by the `AB` and `FB` models in the second stage.

## 6 CONCLUSION & DISCUSSION

In this work, we develop an algorithm for training classifiers that abstain to obtain a favorable fairness guarantee. Simultaneously, we show that our abstaining process incurs much less harm to each individual group's baseline accuracy, compared to existing algorithms. We theoretically analyzed the feasibility of our goal and relate multiple system design parameters to the required abstention rates. We empirically verified the benefits of our proposal. Interesting future directions for our research involve extending our method beyond binary classification tasks to encompass multi-class scenarios. Preliminary considerations suggest transforming the problem into a series of binary classification tasks, with additional design required to refine the flipping mechanism. Another avenue is to include a human subject study, incorporating human annotation for abstained samples into the performance evaluation. Additionally, exploring the reduction of IP constraints could further reduce computational complexity, providing valuable insights for future developments.

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

## A  FAIRNESS NOTION

| Fairness Notion | $\mathscr{D}$ | $\mathscr{E}$ |
|---|---|---|
| Demographic Parity | $\left\| \mathbb{P}(\hat{Y}=1\|Z=z) - \mathbb{P}(\hat{Y}=1\|Z=z') \right\|$ | $\varepsilon$ |
| Equal Opportunity | $\left\| \mathbb{P}(\hat{Y}=1\|Y=1,Z=z) - \mathbb{P}(\hat{Y}=1\|Y=1,Z=z') \right\|$ | $\varepsilon$ |
| Equalized Odds | $\left( \begin{array}{c} \left\| \mathbb{P}(\hat{Y}=1\|Y=1,Z=z) - \mathbb{P}(\hat{Y}=1\|Y=1,Z=z') \right\| \\ \left\| \mathbb{P}(\hat{Y}=0\|Y=0,Z=z) - \mathbb{P}(\hat{Y}=0\|Y=0,Z=z') \right\| \end{array} \right)$ | $\left( \begin{array}{c} \varepsilon \\ \varepsilon \end{array} \right)$ |

Table 3: Fairness notion utilized for Constraint Disparity. We utilize three specific fairness notions, Demographic Parity (DP) (Dwork et al., 2012), Equal Opportunity (EOp) (Hardt et al., 2016), Equalized Odds (EOd) (Hardt et al., 2016).

| Fairness Notion | $\mathscr{D}$ | $\mathscr{E}$ |
|---|---|---|
| Demographic Parity | $\left\| \frac{\sum_{n=1}^{N}\omega_n\cdot 1[\hat{y}_n=1,z_n=z]}{\sum_{i=1}^{N}1[z_n=z]} - \frac{\sum_{n=1}^{N}\omega_n\cdot 1[\hat{y}_n=1,z_n=z']}{\sum_{n=1}^{N}1[z_n=z']} \right\|$ | $\varepsilon$ |
| Equal Opportunity | $\left\| \frac{\sum_{n=1}^{N}\omega_n\cdot 1[\hat{y}_n=1,y_n=1,z_n=z]}{\sum_{i=1}^{N}1[z_n=z,y_n=1]} - \frac{\sum_{n=1}^{N}\omega_n\cdot 1[\hat{y}_n=1,y_n=1,z_n=z']}{\sum_{n=1}^{N}1[z_n=z',y_n=1]} \right\|$ | $\varepsilon$ |
| Equalized Odds | $\left( \begin{array}{c} \left\| \frac{\sum_{n=1}^{N}\omega_n\cdot 1[\hat{y}_n=1,y_n=1,z_n=z]}{\sum_{i=1}^{N}1[z_n=z,y_n=1]} - \frac{\sum_{n=1}^{N}\omega_n\cdot 1[\hat{y}_n=1,y_n=1,z_n=z']}{\sum_{n=1}^{N}1[z_n=z',y_n=1]} \right\| \\ \left\| \frac{\sum_{n=1}^{N}\omega_n\cdot 1[\hat{y}_n=0,y_n=0,z_n=z]}{\sum_{i=1}^{N}1[z_n=z,y_n=0]} - \frac{\sum_{n=1}^{N}\omega_n\cdot 1[\hat{y}_n=0,y_n=0,z_n=z']}{\sum_{n=1}^{N}1[z_n=z',y_n=0]} \right\| \end{array} \right)$ | $\left( \begin{array}{c} \varepsilon \\ \varepsilon \end{array} \right)$ |

Table 4: The emperical version of fairness notions utilized for Constraint Disparity in **IP-Main**.

Tables 3 and 4 display the formulation of three fairness notions we have adopted. It's worth mentioning that all fairness measurements are not conditioned on $h_A$. This means that fairness is measured across the entire dataset, not just for non-abstained samples. We now provide an example of Demographic Parity (Disparity of accept rate) to demonstrate why we take this approach.

**Example A.1.** Consider a group with an acceptance rate of 0.3 from the baseline classifier. If FAN abstains at a rate of 0.1 on samples with negative predictions, our measurement of the acceptance rate, not conditioned on abstentions, should yield a unchanged acceptance rate of 0.3. However, if we condition it on non-abstentions, the new acceptance rate should be $0.3/(0.3+0.6) \approx 0.33$. The former is a more valid measure than the latter, as it takes into account the presence of abstained samples still in the system and the abstention on the negative samples should not impact the accept rate.

## B  DETAILS OF TWO-STAGE TRAINING

Solving Problem **IP-Main** provides us with two $N$-dimensional vectors, namely $\omega$ and $f$. The vector $\omega$ denotes whether to abstain from predicting for every input in the training data, while the vector $f$ represents whether the final prediction $\hat{y}$ needs to be flipped compared to the baseline optimal model $h$. As we have illustrated in Figure 2, we will utilize $\omega$ and $f$ as labels, paired with $X, \hat{Y}$ as training features, to train the Abstention Block $h_A(X, h(X))$ and the Flip Block $h_F(X, h(X))$.

### B.1  ELIMINATING RANDOMNESS OF IP OUTCOMES USING PREDICTION ADJUSTMENT

Although IP provides an optimal solution in terms of $\omega$ and $f$, this section discusses the non-uniqueness of this solution in most cases. As a result, randomness can affect the ability of AB and FB to learn IP decisions effectively.

The input of IP is $z, y, s$. The way IP uses confidence score $s$ is mapping it to $\hat{y}_b = 1[s \geq t_0]$. Note that IP is not using the feature $x$ directly. Instead, the information of $x$ is captured by $\hat{y}_b$. Therefore the IP solution exhibits a certain degree of randomness. Specifically, the IP model does not differentiate between data samples with identical labels and optimal predictions, if they belong to the same group (i.e., $y_1 = y_2, \hat{y}_{b1} = \hat{y}_{b2}, z_1 = z_2$). If the decisions for two samples $x_1$ and $x_2$ are interchanged, the resulting solution would still be optimal for IP. This characteristic can hinder the complete capture of feature information and adversely affect the performance of model training for AB and FB. We provide an illustrative figure for this observation in Figure 7a.

To mitigate the effects of randomness in the IP solution, a Prediction Adjustment (PA) step is incorporated after solving the IP and before the model training, outlined in Algorithm 1 and illurstrated in Figure 6. The core idea of PA is that we abstain from predicting those with the lowest confidence scores predicted by the optimal classifier according to the optimal fraction we learned from the IP solver. Of the remaining, we further select the samples with the lowest confidence scores for flipping (`FB`). Samples with the highest confidence scores, on the other hand, will remain unaffected by this process. This approach is founded on the premise that the highest confidence scores correspond to the most certain predictions made by the

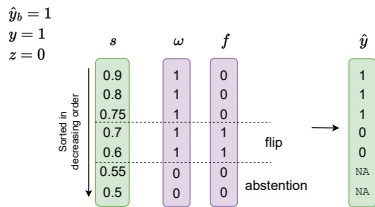

Figure 6: Illustration of Prediction Adjustment.

optimal classifier. Therefore, it is prudent to prioritize abstaining and flipping those with lower confidence scores first. Specifically, for each $(y, \hat{y}_b, z)$ tuples, PA procedure takes the following steps:

- Count the number of individuals that have been abstained, i.e., $n_0 = \sum_i 1[\omega_i = 0]$; the number that has not been abstained but the decision is flipped, i.e., $n_{11} = \sum_n 1[\omega_n = 1, f_n = 1]$.
- Sort the individuals based on their confidence score predicted by baseline optimal $h$.
- Abstain $n_0$ individuals with lowest confidence; for the rest, flip the decision of $n_{11}$ individuals.

---

**Algorithm 1** Prediction Adjustment

---

**Input:** $(x, y, z), \hat{y}_b, h, \omega, f$
**for** $a \in \mathcal{Z}$ **do**
    **for** $y_1 \in \{0, 1\}$ **do**
        **for** $y_2 \in \{0, 1\}$ **do**
            $(\bar{x}, \bar{y}, \bar{w}, \bar{f})$ contains all data samples with $z_n = a, y_n = y_1, \hat{y}_{b,n} = y_2$.
            For all data samples with $\bar{\omega}_n = 0$, set $\bar{f}_n = 0$.
            $n_{11} =$ number of the data samples with $\omega_n = 1, f_n = 1$.
            $n_{10} =$ number of the data samples with $\omega_n = 1, f_n = 0$.
            $n_0 =$ number of the data samples with $\omega_n = 0$.
            Compute the confidence score $h(\bar{x})$.
            Adjust $\omega, f$: With increasing in confidence score, reassign $n_0$ data samples with $\omega_n = 0$, $n_{10}$ samples with $\omega_n = 1, f_n = 0$, $n_{11}$ samples with $\omega_n = 1, f_n = 1$, sequentially.
        **end for**
    **end for**
**end for**
Return $\omega, f$.

---

**Robustness: Prediction Consistency.** The Prediction adjustment technique not only mitigates the randomness introduced by the Iterative Pruning algorithm but also preserves prediction consistency when changes are made to the training data. In practical scenarios, when new data points are added to the training set, sampled from the same distribution $\mathscr{D}$, they are expected to be distributed proportionally across all regions as illustrated in Figure 7b. The Prediction adjustment policy guarantees that the labels of the original data in the training set remain unchanged while incorporating the new data points. For experiment verification of Prediction Consistency see Appendix E.

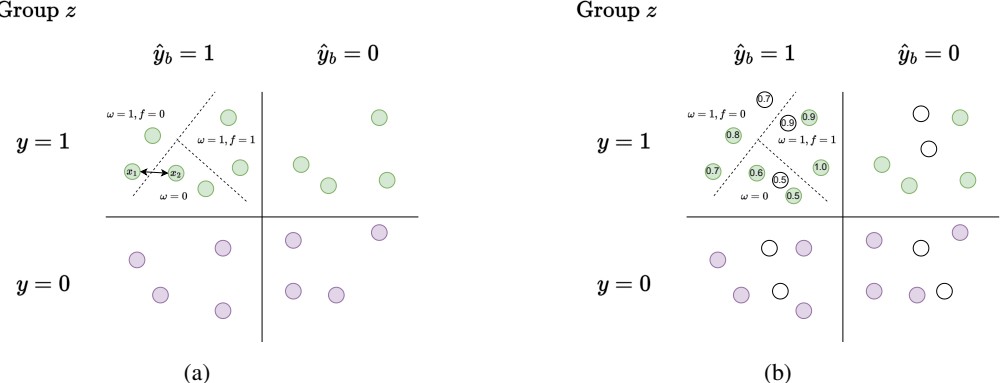

Figure 7: Illustration of the Non-Uniqueness of IP Solutions (a) and prediction consistency Concept (b): (a) The figure depicts the inherent randomness of IP solutions. For a given group $z$, IP can only observe the ground truth label $y$ and the predicted label $\hat{y}_b$. Hence, for instances $x$ with the same $y$ and $\hat{y}_b$, IP cannot differentiate between them. For example, swapping the decisions of $x_1$ and $x_2$ yields a new optimal solution. (b) This figure depicts the concept of prediction consistency, wherein newly sampled data, obtained from the same distribution as the original data, are incorporated into the training set. Given that the newly sampled data are independent and identically distributed (iid), they are expected to be distributed proportionally across all regions. The Prediction adjustment policy is employed to ensure that the labels of the original data remain unchanged.

## B.2   Linear Integer Programming

In Problem **IP-Main**, the presence of quadratic terms in the form of $\hat{y}_n \omega_n$ incurs higher computational costs and renders the problem more difficult to solve than a linear IP. In this section, we present a methodology to transform Problem **IP-Main** into an equivalent linear IP problem. To achieve this, we employ McCormick envelopes and define $u_n = \hat{y}_n \omega_n$. Since $\hat{y}_n$ and $\omega_n$ are binary, the following set of linear constraints can be used to represent $u_n$:

$$u_n = \hat{y}_n \omega_n \Leftrightarrow \begin{cases} u_n \geq 0 \,,\, u_n \leq \omega_n \,,\, u_n \leq h_n \\ (1 - \hat{y}_n)(1 - \omega_n) \geq 0 \Leftrightarrow u_n \geq \hat{y}_n + \omega_n - 1 \end{cases} \tag{5}$$

Intuitively, because we are in the binary setting, we are able to turn a quadratic optimization problem into a linear one. It is important to note that Equation 5 does not introduce any relaxation. Moreover, all the constraints in Equation 5 are linear. Thus, we can replace all the quadratic terms in Problem **IP-Main** with $u_n$ and incorporate the linear constraints specified in Equation 5.

## C   Non-triviality

At the end of Section 4.1, our results indicate that a feasible solution to the IP problem always exists for equal opportunity and equalized odds, regardless of the design parameter values. Notably, our results imply that even when the abstention rate is 0, the IP can solely adjust the flip decisions $f_n$ to satisfy constraints on disparate impact, abstention rate, and no harm. This is because IP has access to the true label $y$.

However, this causes problem. For example, to achieve a "perfect" classifier, IP doesn't abstain from individuals but only flips the decision of $\hat{y}_b$ to make the final outcome exactly equal to $y$, resulting in a $100\%$ accuracy across all groups. Additionally, the true positive rate and true negative rate will both be $100\%$, eliminating any disparities. However, such a "perfect" classifier is trivial. It requires a strong FB to memorize the flipping decision. To eliminate such trivial solutions, a more reasonable approach is to ensure that the IP solution is **no better than** the optimal classifier

$$h_o = \arg\min_{h' \in \mathcal{H}} \mathbb{E}_{\mathcal{D}}[\mathcal{L}(h'(X), Y)], \tag{6}$$

without considering abstentions.

We introduce a non-triviality constraint to the IP with Equal Opportunity as an example:

$$\min_{\omega, f} \quad \textbf{IP-Main} \quad \text{(Equal Opportunity)} \tag{7}$$

$$\text{s.t.a.} \quad \sum_{n=1}^{N} 1[\hat{y}_n \neq y_n, z_n = z] \geq \left( \sum_{n=1}^{N} 1[z_n = z] \right) \cdot e_{o,z}, \forall z \in \mathcal{Z} \quad \text{(Non-triviality)}$$

Here $e_{o,z}$ is the error rate of the optimal classifier for group z. In above s.t.a. stands for "subject to additional" constraints. The non-triviality constraint enforces a realistic requirement that our flipping module alone should not lead to a higher group accuracy compared to the optimal classifier. We prove the following theorem:

**Theorem C.1.** *When the baseline is optimal, i.e., $h = h_o$, if for all $z \in \mathcal{Z}$, $e_z \leq 1 - \tau_z$, $\tau_z(1 - \varepsilon) + (1 - e_z)(1 - \delta_z) \leq 1$, and $\eta_z = 0$, problem 7 is feasible.*

# D PROOF

## D.1 NOTATIONS

For simplicity, we introduce some notations that will be used in the proof. Figure 8 illustrates the distribution ratios of different regions of two groups.

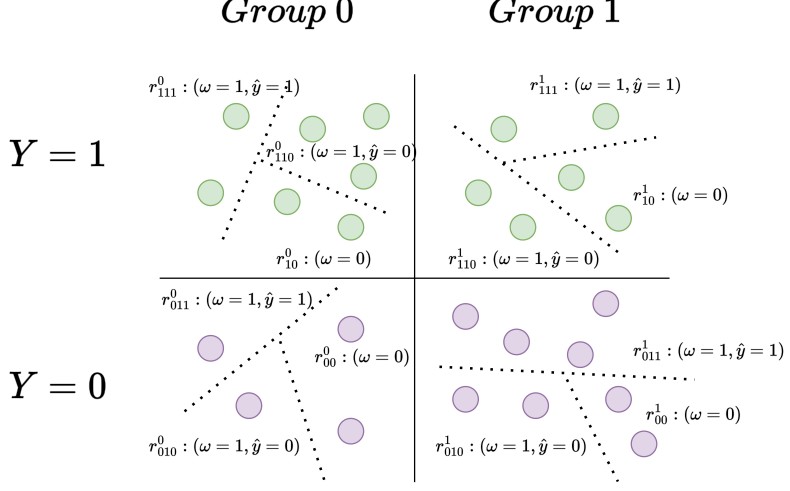

Figure 8: Division of individuals into four regions based on their sensitive attribute and label. The ratio, denoted as $r^z ijk$, represents the proportion of individuals of group $z$ with specific combinations of $y = i$, $\omega = j$, and $\hat{y} = k$. For $\omega = 0$, define $r_{i0}^z = r_{i00}^z + r_{i01}^z$. By definition, $r_{111}^z + r_{110}^z + r_{10}^z + r_{011}^z + r_{010}^z + r_{00}^z = 1$ holds true, ensuring the sum of all ratios within group $z$ is equal to one. Define $r^z = \{r_{111}^z, r_{110}^z, r_{10}^z, r_{011}^z, r_{010}^z, r_{00}^z\}$.

Note that $e_z' = (1 + \eta_z)e_z$. For notation simplicity, define $a_z' = 1 - e_z'$.

## D.2 PROOF OF LEMMA D.1

**Lemma D.1.** *Under fairness notion Demographic Parity or Equal Opportunity, if Problem **IP-Main** is feasible, there must exist a solution such that the resulting classifier does not abstain any samples with label 0, i.e., for all $n$ such that $y_n = 0$, it holds that $\omega_n = 1$.*

Lemma D.1 indicates abstaining data samples with label 0 are not required by IP.

*Proof.* Without loss of generality, we consider group 0. Since Problem **IP-Main** is feasible, there exist a $r^0$ that Constraints **Abstention Rate**, **Disparity**, **No Harm** hold.

Constraint No Harm can be written as

$$\frac{r_{110}^0 + r_{011}^0}{r_{111}^0 + r_{110}^0 + r_{011}^0 + r_{010}^0} \le e_0' \Leftrightarrow a_0'(r_{110}^0 + r_{011}^0) \le e_0'(r_{111}^0 + r_{010}^0) \tag{8}$$

Constraint Abstention Rate can be written as

$$r_{10}^0 + r_{00}^0 \le \delta_0 \tag{9}$$

For Demographic Parity, constraint Disparity indicates

$$r_{111}^z + r_{011}^z - \varepsilon \le r_{111}^0 + r_{011}^0 \le r_{111}^z + r_{011}^z + \varepsilon, \forall z \tag{10}$$

If the feasible solution $r_{00}^0 > 0$, define $r_{010}^{0\prime} = r_{010}^0 + r_{00}^0, r_{00}^{0\prime} = 0$, then it's not hard to verify that constraints 8, 9 and 10 still hold for $r_{010}^{0\prime}$ and $r_{00}^{0\prime} = 0$. $r_{00}^{0\prime} = 0$ indicates abstaining no individual with label 0.

For Equal Opportunity, constraint Disparity indicates

$$\frac{r_{111}^z}{\tau_z} - \varepsilon \le \frac{r_{111}^0}{\tau_0} \le \frac{r_{111}^z}{\tau_z} + \varepsilon, \forall z \tag{11}$$

Similarly, if the feasible solution $r_{00}^0 > 0$, define $r_{010}^{0\prime} = r_{010}^0 + r_{00}^0, r_{00}^{0\prime} = 0$, then it's not hard to verify that constraints 8, 9 and 11 still hold for $r_{010}^{0\prime}$ and $r_{00}^{0\prime} = 0$. $r_{00}^{0\prime} = 0$ indicates abstaining no individual with label 0.

### D.3 PROOF OF THEOREM 4.1

By Lemma D.1, we have Problem **IP-Main** is feasible iff there exist a solution that $r_{00}^z = 0, \forall z$.

For any two group $z = 0, 1$, without loss of generality, assume $\tau_1 \le \tau_0$. Define $\Delta\tau = \tau_0 - \tau_1$.

Similar to 9, 8 and 10, the constraint of **IP-Main** can be written as

$$\begin{cases} r_{111}^0 - r_{010}^0 - \Delta\tau - \varepsilon \le r_{111}^1 - r_{010}^1 \\ r_{111}^0 - r_{010}^0 - \Delta\tau + \varepsilon \ge r_{111}^1 - r_{010}^1 \\ \qquad\qquad\qquad\qquad r_{10}^1 \le \delta_1 \\ \qquad\qquad\qquad\qquad r_{10}^0 \le \delta_0 \\ a_0'(r_{110}^0 + r_{011}^0) \le e_0'(r_{111}^0 + r_{010}^0) \\ a_1'(r_{110}^1 + r_{011}^1) \le e_1'(r_{111}^1 + r_{010}^1) , \\ \qquad\qquad r_{111}^1 + r_{10}^1 + r_{110}^1 = \tau_1 \\ \qquad\qquad r_{111}^0 + r_{10}^0 + r_{110}^0 = \tau_0 \\ \qquad\qquad\qquad r_{010}^1 + r_{011}^1 = 1 - \tau_1 \\ \qquad\qquad\qquad r_{010}^0 + r_{011}^0 = 1 - \tau_0 \\ \qquad\qquad\qquad\qquad r^1, r^0 \ge \mathbf{0} \end{cases} \tag{12}$$

Let $r_{011}^1 = 1 - \tau_1 - r_{010}^1, r_{011}^0 = 1 - \tau_0 - r_{010}^0, r_{110}^1 = \tau_1 - r_{111}^1 - r_{10}^1, r_{110}^0 = \tau_0 - r_{111}^0 - r_{10}^0$, 12 becomes

$$\begin{cases} r^0_{111} - r^0_{010} - \Delta\tau - \varepsilon \le r^1_{111} - r^1_{010} \\ r^0_{111} - r^0_{010} - \Delta\tau + \varepsilon \ge r^1_{111} - r^1_{010} \\ r^1_{10} \le \delta_1 \\ r^0_{10} \le \delta_0 \\ r^1_{111} + a'_1 r^1_{10} + r^1_{010} \ge a'_1 \\ r^0_{111} + a'_0 r^0_{10} + r^0_{010} \ge a'_0 \\ r^1_{111} + r^1_{10} \le \tau_1 \\ r^0_{111} + r^0_{10} \le \tau_0 \\ r^1_{010} \le 1 - \tau_1 \\ r^0_{010} \le 1 - \tau_0 \\ r^1_{111}, r^1_{10}, r^1_{010}, r^0_{111}, r^0_{10}, r^0_{010} \ge 0 \end{cases} \tag{13}$$

Extract the condition of $r^1_{10}$ from 13, we have

$$\begin{cases} 0 \le r^1_{10} \le \delta_1 \\ r^1_{10} \ge \dfrac{a'_1 - r^1_{010} - r^1_{111}}{a'_1} \\ r^1_{10} \le \tau_1 - r^1_{111} \end{cases} \tag{14}$$

To let 14 feasible, the following additional inequalities need to be added to 13,

$$\begin{cases} \dfrac{a'_1 - r^1_{010} - r^1_{111}}{a'_1} \le \delta_1 \\ \dfrac{a'_1 - r^1_{010} - r^1_{111}}{a'_1} \le \tau_1 - r^1_{111} \\ 0 \le \tau_1 - r^1_{111} \end{cases} \tag{15}$$

Similar requirements hold for group 0. Thus,

$$\text{13 is feasible} \Leftrightarrow \begin{cases} r^0_{111} - r^0_{010} - \Delta\tau - \varepsilon \le r^1_{111} - r^1_{010} \\ r^0_{111} - r^0_{010} - \Delta\tau + \varepsilon \ge r^1_{111} - r^1_{010} \\ r^1_{010} \le 1 - \tau_1 \\ r^0_{010} \le 1 - \tau_0 \\ r^0_{010} + r^0_{111} \ge a'_0(1 - \delta_0) \\ e'_0 r^0_{111} + r^0_{010} \ge a'_0(1 - \tau_0) \\ r^0_{111} \le \tau_0 \\ r^1_{010} + r^1_{111} \ge a'_1(1 - \delta_1) \\ e'_1 r^1_{111} + r^1_{010} \ge a'_1(1 - \tau_1) \\ r^1_{111} \le \tau_1 \\ r^1_{111}, r^1_{010}, r^0_{111}, r^0_{010} \ge 0 \end{cases} \tag{16}$$

Further extract conditions of $r^0_{010}$, we have

$$\text{13 is feasible} \Leftrightarrow \begin{cases} r_{010}^1 \leq 1 - \tau_1 \\ r_{111}^0 \leq \tau_0 \\ r_{010}^1 + r_{111}^1 \geq a_1'(1 - \delta_1) \\ e_1' r_{111}^1 + r_{010}^1 \geq a_1'(1 - \tau_1) \\ r_{111}^1 \leq \tau_1 \\ r_{111}^0 \leq 1 - \tau_1 + r_{111}^1 - r_{010}^1 + \varepsilon \\ 2r_{111}^0 \geq a_0'(1 - \delta_0) + \Delta\tau - \varepsilon + r_{111}^1 - r_{010}^1 \\ r_{111}^0 \geq a_0'(1 - \delta_0) + \tau_0 - 1 \\ (1 + e_0')r_{111}^0 \geq a_0'(1 - \tau_0) + \Delta\tau - \varepsilon + r_{111}^1 - r_{010}^1 \\ r_{111}^0 \geq \Delta\tau - \varepsilon + r_{111}^1 - r_{010}^1 \\ r_{111}^1, r_{010}^1, r_{111}^0 \geq 0 \end{cases}, \qquad (17)$$

Extract conditions of $r_{111}^0$, we have

$$\text{13 is feasible} \Leftrightarrow \begin{cases} r_{010}^1 \leq 1 - \tau_1 \\ r_{010}^1 + r_{111}^1 \geq a_1'(1 - \delta_1) \\ e_1' r_{111}^1 + r_{010}^1 \geq a_1'(1 - \tau_1) \\ r_{111}^1 \leq \tau_1 \\ r_{111}^1 - r_{010}^1 \leq \tau_1 + \varepsilon \\ r_{111}^1 - r_{010}^1 \leq \tau_0 + \tau_1 + \varepsilon - a_0' \\ \tau_0 + \tau_1 - (\dfrac{2}{e_0'} + 1)\varepsilon - 2 \leq r_{111}^1 - r_{010}^1 \\ r_{111}^1 - r_{010}^1 \geq a_0'(1 - \delta_0) + \tau_0 + \tau_1 - 2 - \varepsilon \\ r_{111}^1, r_{010}^1 \geq 0 \end{cases}, \qquad (18)$$

Define $r_+^1 = r_{111}^1 + r_{010}^1, r_-^1 = r_{111}^1 - r_{010}^1$, then we have $0 \leq r_+^1 \leq 1, \tau_1 - 1 \leq r_-^1 \leq \tau_1$.

$$\text{13 is feasible} \Leftrightarrow \begin{cases} r_+^1 \geq a_1'(1 - \delta_1) \\ r_-^1 \geq r_+^1 + 2\tau_1 - 2 \\ a_1' r_-^1 \leq (1 + e_1')r_+^1 - 2a_1'(1 - \tau_1) \\ r_-^1 \leq 2\tau_1 - r_+^1 \\ r_-^1 \leq \tau_0 + \tau_1 + \varepsilon - a_0' \\ r_-^1 \geq \tau_0 + \tau_1 - (\dfrac{2}{e_0'} + 1)\varepsilon - 2 \\ r_-^1 \geq a_0'(1 - \delta_0) + \tau_0 + \tau_1 - 2 - \varepsilon \\ r_-^1 \leq r_+^1 \\ r_-^1 \geq -r_+^1 \end{cases}, \qquad (19)$$

Extract conditions of $r_-^1$, we have

$$13 \text{ is feasible} \Leftrightarrow \begin{cases} r_+^1 \geq 0 \\ r_+^1 \geq a_0'(1-\delta_0) + \tau_0 + \tau_1 - 2 - \varepsilon \\ \left(1 + \dfrac{2e_1'}{a_1'}\right) r_+^1 \geq a_0'(1-\delta_0) + \Delta\tau - \varepsilon \\ r_+^1 \geq a_1'(1-\tau_1) \\ r_+^1 \geq a_0' - \tau_0 - \tau_1 - \varepsilon \\ r_+^1 \leq 1 \\ r_+^1 \leq 2 + \varepsilon - a_0'(1-\delta_0) - \Delta\tau \end{cases}, \tag{20}$$

$$13 \text{ is feasible} \Leftrightarrow a_0'(1-\delta_0) + \Delta\tau \leq 1 + \varepsilon + e_1', \tag{21}$$

Thus, if for any two groups $z, z' \in \mathcal{Z}$ such that $\tau_z \geq \tau_{z'}$, $a_z'(1-\delta_z) + \tau_z - \tau_{z'} \leq 1 + \varepsilon + e_{z'}'$ holds, then Problem **IP-Main** is feasible.

### D.4 PROOF OF THEOREM 4.3

*Proof.* Similarly, for any two groups $0, 1$, the constraints of Problem **IP-Main** under Equal Opportunity are

$$\begin{cases} \dfrac{r_{111}^0}{\tau_0} - \varepsilon \leq \dfrac{r_{111}^1}{\tau_1} \leq \dfrac{r_{111}^0}{\tau_0} + \varepsilon \\ r_{10}^1 \leq \delta_1 \\ r_{10}^0 \leq \delta_0 \\ \tau_1 - e_1' \leq r_{111}^1 + a_1' r_{10}^1 \\ \tau_0 - e_0' \leq r_{111}^0 + a_0' r_{10}^0 \\ r_{111}^1 + r_{10}^1 \leq \tau_1 \\ r_{111}^0 + r_{10}^0 \leq \tau_0 \\ r_{111}^1, r_{10}^1, r_{111}^0, r_{10}^0 \geq 0 \end{cases} \tag{22}$$

Extract the conditions of $r_{10}^1$ and $r_{10}^0$, we have the following equivalent constraints:

$$\begin{cases} \dfrac{\tau_1}{\tau_0} r_{111}^0 \leq \tau_1(1+\varepsilon) \\ \dfrac{\tau_0}{\tau_1} r_{111}^1 \leq \tau_0(1+\varepsilon) \\ \dfrac{\tau_1}{\tau_0} r_{111}^0 \geq \tau_1 - e_1' - a_1'\delta - \varepsilon\tau_1 \\ \dfrac{\tau_0}{\tau_1} r_{111}^1 \geq \tau_0 - e_0' - a_0'\delta - \varepsilon\tau_0 \\ 0 \leq r_{111}^0 \leq \tau_0 \\ 0 \leq r_{111}^1 \leq \tau_1 \end{cases} \tag{23}$$

To let the above feasible, we need

$$\begin{cases} \dfrac{(\tau_1 - e_1' - a_1'\delta - \varepsilon\tau_1)\tau_0}{\tau_1} \leq \tau_0 \\ \dfrac{(\tau_0 - e_0' - a_0'\delta - \varepsilon\tau_0)\tau_1}{\tau_0} \leq \tau_1 \end{cases} \tag{24}$$

24 always holds. Thus, Problem **IP-Main** always feasible under Equal Opportunity.

## D.5 Proof of Theorem 4.4

*Proof.* Under Equalized Odds, for any group $z$, let $r^z_{111} = \tau_z, r^z_{110} = 0, r^z_{10} = 0, r^z_{011} = 0, r^z_{010} = 0, r^z_{00} = 0$, then we can verify that all the constraints of Problem **IP-Main** hold.

## D.6 Proof of Theorem C.1

For any group 1, the constraints of Problem 7 are

$$
\begin{cases}
\dfrac{r^0_{111}}{\tau_0} - \varepsilon \le \dfrac{r^1_{111}}{\tau_1} \le \dfrac{r^0_{111}}{\tau_0} + \varepsilon \\
r^1_{101} + r^1_{100} + r^1_{000} + r^1_{001} \le \delta_1 \\
\dfrac{r^1_{110} + r^1_{011}}{r^1_{110} + r^1_{111} + r^1_{011} + r^1_{010}} \le e'_1 \\
r^1_{110} + r^1_{011} + r^1_{100} + r^1_{001} \ge e_1 \\
r^1_{110} + r^1_{111} + r^1_{101} + r^1_{100} = \tau_1 \\
r^1_{011} + r^1_{010} + r^1_{000} + r^1_{001} = 1 - \tau_1 \\
r^1 \ge 0
\end{cases}
\tag{25}
$$

where group 0 represents any group other than group 1.

If 25 are feasible, there must exists a solution such that $r^1_{101} = r^1_{000} = 0$. Here we provide proof for $r^1_{101} = 0$. Note that the proof of $r^1_{000} = 0$ is similar.

If $r^1_{101} = x > 0$, we can easily adjust $r^1_{100}$ increase by $x$, $r^1_{101} = 0$ so that the new solution also satisfy 25.

Thus, let $r^1_{101} = r^1_{000} = 0$ and $r^1_{100} = \tau_1 - r^1_{110} - r^1_{111}, r^1_{001} = 1 - \tau_1 - r^1_{011} - r^1_{010}$, Problem 7 holds iff

$$
\begin{cases}
\dfrac{r^0_{111}}{\tau_0} - \varepsilon \le \dfrac{r^1_{111}}{\tau_1} \le \dfrac{r^0_{111}}{\tau_0} + \varepsilon \\
r^1_{110} + r^1_{011} + r^1_{111} + r^1_{010} \ge 1 - \delta_1 \\
a'_1 \left( r^1_{110} + r^1_{011} \right) \le e'_1 \left( r^1_{111} + r^1_{010} \right) \\
r^1_{111} + r^1_{010} \le a_1 \\
r^1_{110} + r^1_{111} \le \tau_1 \\
r^1_{011} + r^1_{010} \le 1 - \tau_1
\end{cases}
\tag{26}
$$

Using similar method in the proof of Theorem 4.1 to solve 26.

## D.7 Proof of Theorem 4.5

Note that when the equal abstention rate constraints are added, Lemma D.1 still holds. The reason is $r^1_{00} = r^0_{00} = 0$ already yields equal abstention rate, since the abstention rates of individuals with negative label are both 0.

Similar to 12, the constraints of Problem 3 can be written as

$$
\begin{cases}
r_{111}^0 - r_{010}^0 - \Delta\tau - \varepsilon \leq r_{111}^1 - r_{010}^1 \\
r_{111}^0 - r_{010}^0 - \Delta\tau + \varepsilon \geq r_{111}^1 - r_{010}^1 \\
\qquad\qquad\qquad\qquad\qquad r_{10}^1 \leq \delta_1 \\
\qquad\qquad\qquad\qquad\qquad r_{10}^0 \leq \delta_0 \\
\qquad\qquad r_{10}^1 \leq r_{10}^0 \dfrac{\tau_1}{\tau_0} + \tau_1\sigma_1 \\
\qquad\qquad r_{10}^1 \geq r_{10}^0 \dfrac{\tau_1}{\tau_0} - \tau_1\sigma_1 \\
a_0'(r_{110}^0 + r_{011}^0) \leq e_0'(r_{111}^0 + r_{010}^0) \\
a_1'(r_{110}^1 + r_{011}^1) \leq e_1'(r_{111}^1 + r_{010}^1) \\
\qquad\qquad r_{111}^1 + r_{10}^1 + r_{110}^1 = \tau_1 \\
\qquad\qquad r_{111}^0 + r_{10}^0 + r_{110}^0 = \tau_0 \\
\qquad\qquad\qquad r_{010}^1 + r_{011}^1 = 1 - \tau_1 \\
\qquad\qquad\qquad r_{010}^0 + r_{011}^0 = 1 - \tau_0 \\
\qquad\qquad\qquad\qquad\qquad r^1, r^0 \geq \mathbf{0}
\end{cases}
, \tag{27}
$$

Compare to 12, the additional constraints of $r_{10}^1$ are $r_{10}^0 \frac{\tau_1}{\tau_0} - \tau_1\sigma_1 \leq r_{10}^1 \leq r_{10}^0 \frac{\tau_1}{\tau_0} + \tau_1\sigma_1$. Plug in them and extract the condition of $r_{10}^1$ yields

$$
\begin{cases}
\qquad\qquad 0 \leq r_{10}^1 \leq \delta_1 \\
\qquad\qquad 0 \leq r_{10}^0 \leq \delta_0 \\
r_{10}^1 \geq \dfrac{a_1' - r_{010}^1 - r_{111}^1}{a_1'} \\
r_{10}^0 \geq \dfrac{a_0' - r_{010}^0 - r_{111}^0}{a_0'} \\
\qquad\qquad r_{10}^1 \leq \tau_1 - r_{111}^1 \\
\qquad\qquad r_{10}^0 \leq \tau_0 - r_{111}^0 \\
\tau_1 r_{10}^0 \leq \tau_0 r_{10}^1 + \tau_0\tau_1\sigma_1 \\
\tau_1 r_{10}^0 \geq \tau_0 r_{10}^1 - \tau_0\tau_1\sigma_1
\end{cases}
, \tag{28}
$$

The last two inequalities are the additional constraints introduced by equal abstention rate constraints.

Thus, if $\tau_0\sigma_1 \geq \delta_0$ and $\delta_1 \leq \tau_1\sigma_1$ hold, then the feasibility of Problem 3 will reduce to the non equal abstention rate case **IP-Main** (Equation 16). Therefore, a sufficient condition of Problem 3 is

$$
\delta_1 \leq \tau_1\sigma_1, \quad \delta_0 \leq \tau_0\sigma_1, \quad \delta_0 \geq 1 - \frac{1 + \varepsilon + (1+\eta_1)e_1 - \tau_0 + \tau_1}{1 - (1+\eta_0)e_0}. \tag{29}
$$

## D.8 EQUAL ABSTENTION RATE UNDER EQUAL OPPORTUNITY AND EQUALIZED ODDS

**Theorem D.2.** *Problem 3 is always feasible under Equal Opportunity (Equalized Odds).*

This theorem holds because Problem **IP-Main** is always feasible even when all the groups have abstention rate 0 for the individuals with either positive or negative labels. In this case, the abstention rates are already equal.

## D.9 Worse Performance under Equal Abstention Rate

**Theorem D.3.** *(Worse Performance) Let $\omega^{(1)}, f^{(1)}$ be the result of Problem **IP-Main**, let $\omega^{(2)}, f^{(2)}$ be the result of Problem 3, then*

$$\sum_{n=1}^{N} \omega_n^{(1)} \cdot 1[\hat{y}_n^{(1)} \neq y_n] \leq \sum_{n=1}^{N} \omega_n^{(2)} \cdot 1[\hat{y}_n^{(2)} \neq y_n]$$

$$\frac{\sum_{n=1}^{N} \omega_n^{(1)} \cdot 1[\hat{y}_n^{(1)} \neq y_n, z_n = z]}{\sum_{n=1}^{N} \omega_n^{(1)} 1[z_n = z]} \leq \frac{\sum_{n=1}^{N} \omega_n^{(2)} \cdot 1[\hat{y}_n^{(2)} \neq y_n, z_n = z]}{\sum_{n=1}^{N} \omega_n^{(2)} 1[z_n = z]}, \forall z \in \mathcal{Z}$$

$$\bar{\mathscr{D}}(f^{(1)}, \omega^{(1)}) \leq \bar{\mathscr{D}}(f^{(2)}, \omega^{(2)}) \tag{30}$$

## E  Experiments

| Dataset | Adult | Compas | Law |
|---------|-------|--------|-----|
| Train data | 37969 | 8467 | 12928 |
| Val data | 7911 | 1765 | 2694 |
| Test data | 9888 | 2206 | 3368 |

Table 5: Size of train, val, test data of each dataset.

**Dataset.**  We utilized three real-world datasets, namely `Adult` Dua & Graff (2017), `Compas` Bellamy et al. (2018), and `Law` Bellamy et al. (2018). The sizes of the training, validation, and test data can be found in Table 5. The task of the `Adult` dataset is to predict whether a person's income exceeds \$50k per year based on various demographic and employment-related features. The sensitive attribute is `sex`, with group 1 representing `male` and group 2 representing `female`. In the `Compas` dataset, the task is to predict the likelihood of a defendant committing a future crime, aiming to assist judges in making more informed decisions about pretrial detention and release. The task has been categorized, and the sensitive attribute is `race`, with group 1 representing `Caucasian` and group 2 representing `African-American`. The `Law` dataset aims to predict the likelihood of a law school student dropping out within the first two years based on various academic and demographic attributes. The task has been categorized, and the sensitive attribute is `race`, with group 1 representing `White` and group 0 representing `Black`.

**Neural Network.**  To train the models, namely `Baseline Optimal`, `AB`, and `FB`, we utilized a Multi-Layer Perceptron (MLP) neural network architecture implemented in PyTorch. The architecture configuration for the `Adult` dataset consists of two layers, each with a dimension of 300. For the `Compas` and `Law` datasets, we employed two layers, each with a dimension of 100. A dropout layer with a dropout probability of 0.5 was applied between the two hidden layers. The Rectified Linear Unit (ReLU) function was used as the activation function. We run the experiments on a single T100 GPU.

### E.1  Baseline Optimal

Table 6 shows the performance of the baseline optimal model on both the training and test datasets.

| | Adult | Compas | Law |
|---|---|---|---|
| Accuracy (%) (Overall) | 92.08 (89.11) | 72.33 (70.31) | 82.86 (81.23) |
| Accuracy (%) (Group 1) | 98.81 (96.67) | 68.99 (66.11) | 82.04 (81.04) |
| Accuracy (%) (Group 2) | 90.28 (87.07) | 75.70 (74.33) | 84.21 (81.57) |
| Disparity (DP) | 0.59 (0.61) | 0.19 (0.19) | 0.16 (0.13) |
| Disparity (EOp) | 0.05 (0.20) | 0.26 (0.22) | 0.11 (0.09) |
| Disparity (EOd) | 0.13 (0.23) | 0.18 (0.18) | 0.08 (0.08) |

Table 6: Performance of the Baseline Optimal Classifier. (Values in parentheses represent results on test data.)

### E.2 OVERALL PERFORMANCE

The performance comparison of `FAN` with `LTD`, `FSCS` on each dataset are shown in this section. We preform 5 runs for `FAN`, under each setting. In most case, `FAN` achieves best performance on both accuracy and disparity.

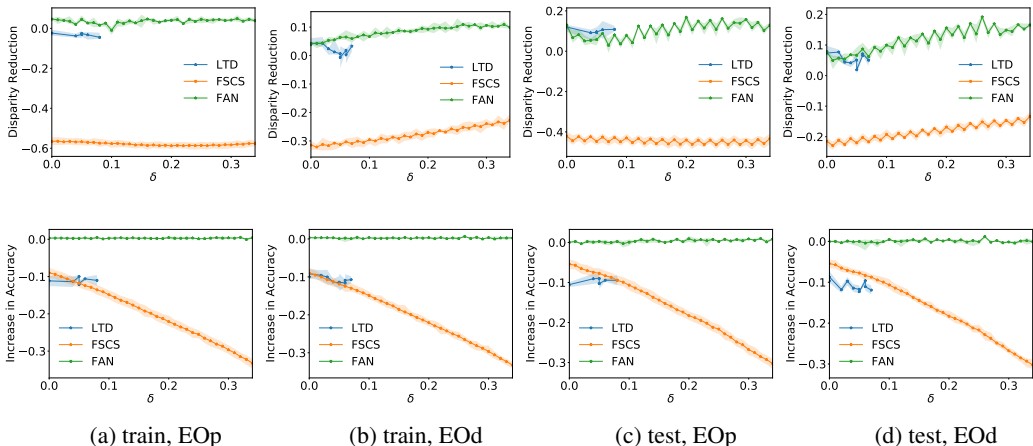

| (a) train, EOp | (b) train, EOd | (c) test, EOp | (d) test, EOd |

Figure 9: Comparison of `FAN` with baseline algorithms on `Adult`. The first row shows the disparity reduction ( compared to baseline optimal), while the second row shows the minimum increase in group accuracy compared to baseline optimal. For `FAN`, $\eta_z$ is set to 0, i.e., no tolerance for reducing accuracy.

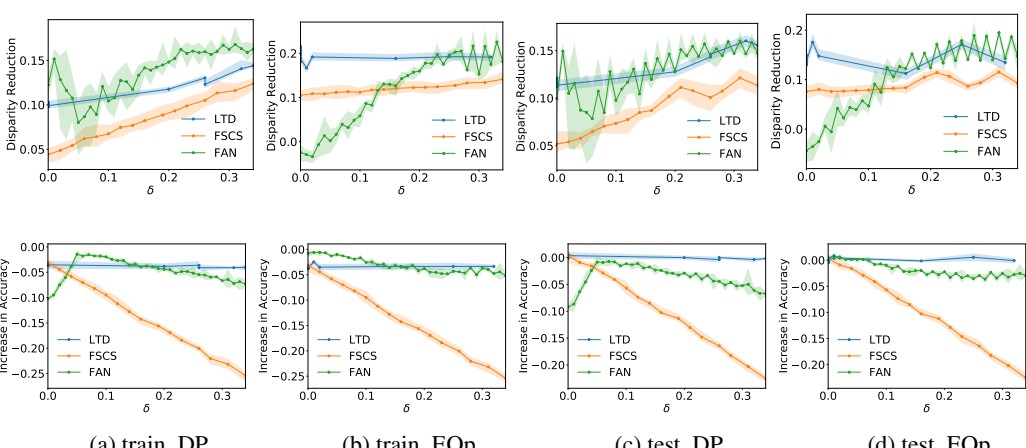

| (a) train, DP | (b) train, EOp | (c) test, DP | (d) test, EOp |

Figure 10: Comparison of `FAN` with baseline algorithms on `Compas`. The first row shows the disparity reduction ( compared to baseline optimal), while the second row shows the minimum group accuracy increase compared to baseline optimal. For `FAN`, $\eta_z$ is set to 0, i.e., no tolerance for reducing accuracy.

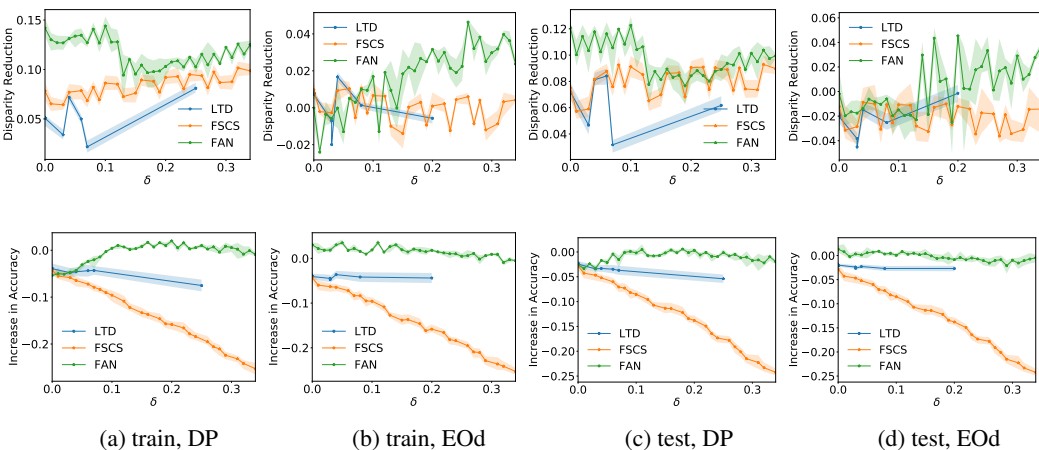

(a) train, DP       (b) train, EOd       (c) test, DP       (d) test, EOd

Figure 11: Comparison of FAN with baseline algorithms on Law. The first row shows the disparity reduction ( compared to baseline optimal), while the second row shows the minimum group accuracy increase compared to baseline optimal. For FAN, $\eta_z$ is set to $0$, i.e., no tolerance for reducing accuracy.

### E.3 COMPARE TO BASELINE OPTIMAL CLASSIFIER

In this section, FAN is compared with baseline optimal classifier under various abstention rate and disparity tolerance.

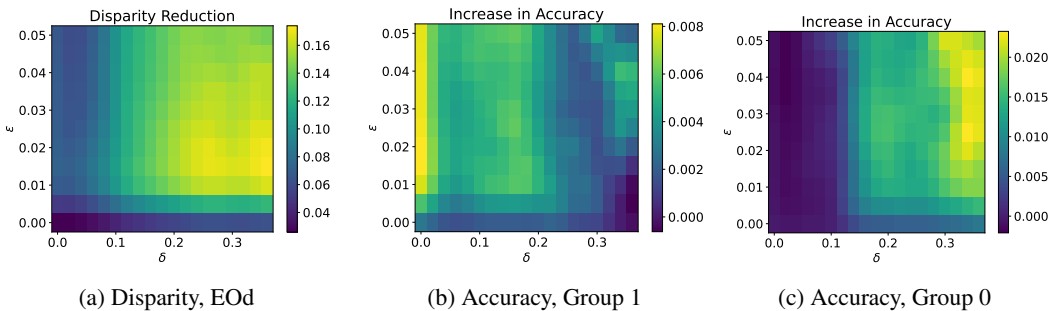

(a) Disparity, EOd       (b) Accuracy, Group 1       (c) Accuracy, Group 0

Figure 12: Disparity reduction and increased accuracy for each group performed on Adult, compared to baseline optimal classifier. This plot illustrates the performance on the testing data. (a) demonstrates the disparity reduction in terms of Equalized odds, while (b) and (c) showcase the increases in accuracy for group 1 and group 0, separately. The x-axis represents the maximum permissible abstention rate, while the y-axis represents the maximum allowable disparity.

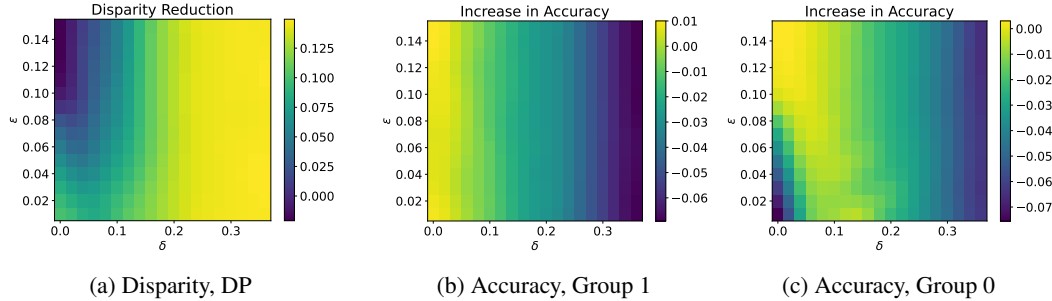

(a) Disparity, DP    (b) Accuracy, Group 1    (c) Accuracy, Group 0

Figure 13: Disparity reduction and increased accuracy for each group performed on `Compas`, compared to baseline optimal classifier. This plot illustrates the performance on the testing data. (a) demonstrates the disparity reduction in terms of Demographic Parity, while (b) and (c) showcase the increases in accuracy for group 1 and group 0, separately. The x-axis represents the maximum permissible abstention rate, while the y-axis represents the maximum allowable disparity.

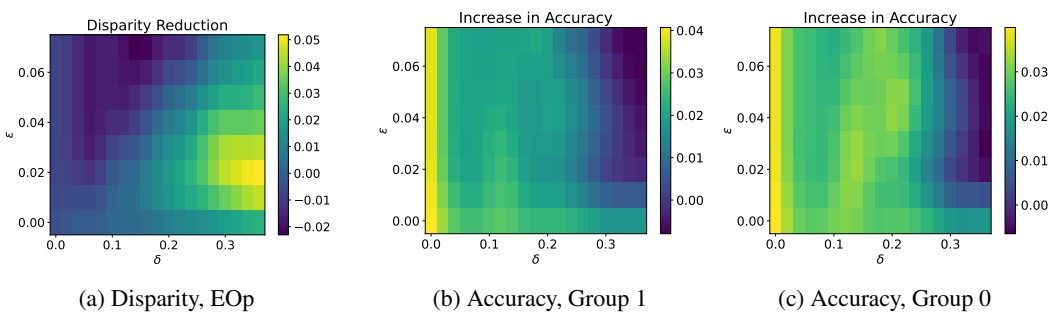

(a) Disparity, EOp    (b) Accuracy, Group 1    (c) Accuracy, Group 0

Figure 14: Disparity reduction and increased accuracy for each group performed on `Law`, compared to baseline optimal classifier. This plot illustrates the performance on the testing data. (a) demonstrates the disparity reduction in terms of Equal Opportunity, while (b) and (c) showcase the increases in accuracy for group 1 and group 0, separately. The x-axis represents the maximum permissible abstention rate, while the y-axis represents the maximum allowable disparity.

### E.4 IMPACT OF $\eta$

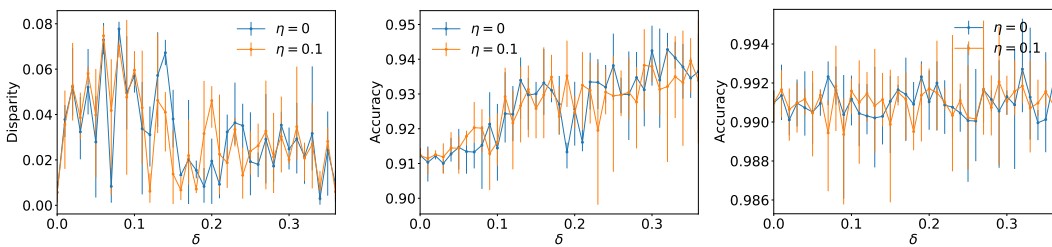

Figure 15: Impact of $\eta$. This experiment is performed on `Adult` under Equal Opportunity, on training data. $\eta$ takes $0, 0.1$. The left shows the disparity of under each setting; the middle shows the accuracy of group 1; the right shows the accuracy of group 0. Relax the error rate control yields similar result, since the objective of IP will still encourage the system to be more accurate.

### E.5 PREDICTION CONSISTENCY

To demonstrate the consistency of IP predictions, which involves abstaining and flipping decisions for individuals with identical features within the same group, we duplicated $\frac{1}{5}$ of the data across all

datasets. This duplication serves to illustrate that IP make the same prediction to these duplicated instances. Details referred to Table 7.

| Dataset | Adult (EO) | Adult (EOs) | Compas (DP) | Compas (EO) | Law (DP) | Law (EO) |
|---|---|---|---|---|---|---|
| Consistent rate ($\omega$) | 0.99 | 0.99 | 1.0 | 1.0 | 0.99 | 0.99 |
| Consistent rate ($f$) | 1.0 | 0.99 | 1.0 | 1.0 | 0.99 | 1.0 |

Table 7: Prediction Consistency Evaluation on IP.

### E.6 STAGE II: SURROGATE MODEL TRAINING LOSS

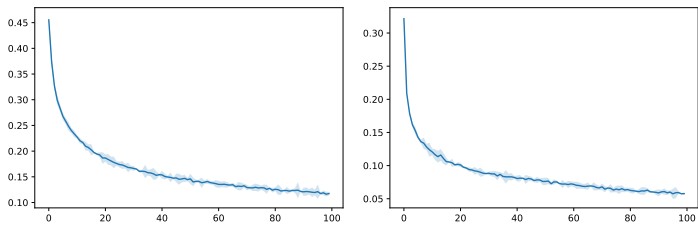

Figure 16: Loss of AB (a) and FB (b), performed on Adult, under Demographic parity. $\delta = 0.1, \varepsilon = 0.02$. We perform 5 runs and plot the average and standard deviation. The corresponding average accuracy is 92.20% and 97.79%.

### E.7 FEASIBILITY REGION

We perform experiments to verify the correctness of Theorem 4.1.

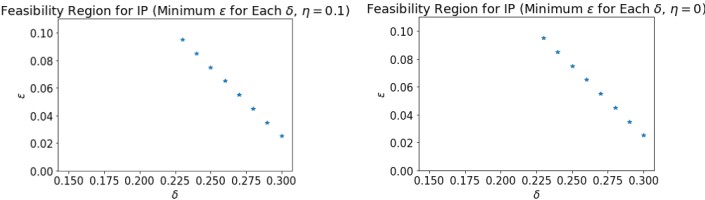

Figure 17: Minimum $\varepsilon$ allowed for each $\delta$, performed on Adult, under Demographic parity. The left figure shows the result under $\eta = 0.1$, while the right figure shows the result under $\eta = 0$. Both figures show the linearity proved in Theorem 4.1.

### E.8 TIME COST

| | Adult | | | Compas | | | Law | | |
|---|---|---|---|---|---|---|---|---|---|
| | DP | EO | EOd | DP | EO | EOd | DP | EO | EOd |
| IP | 36.76 | 22.05 | 33.85 | 5.02 | 4.92 | 10.74 | 12.58 | 15.95 | 16.81 |
| | (1.08) | (1.25) | (1.58) | (1.53) | (0.92) | (0.61) | (0.84) | (0.79) | (0.93) |
| AB | 14.33 | 20.06 | 18.39 | 7.24 | 7.12 | 14.16 | 27.68 | 28.99 | 64.04 |
| | (1.02) | (0.98) | (1.14) | (1.67) | (0.89) | (0.73) | (0.97) | (0.84) | (1.20) |
| FB | 14.72 | 9.59 | 22.05 | 6.90 | 9.88 | 10.52 | 28.00 | 26.82 | 41.87 |
| | (1.65) | (0.72) | (0.96) | (1.31) | (0.75) | (0.65) | (0.81) | (0.87) | (0.88) |

Table 8: Average time cost (in seconds) of solving IP, training AB and FB, respectively. The experiments run on a single T100 GPU. The numbers in parentheses are the corresponding std.

## F  HUMAN ANNOTATION

In the case that human annotation exists for training data, our method can be applied with some modifications of the IP. For example, under Demographic Parity,

$$
\min_{\omega,f} \quad \sum_{n=1}^{N} 1[\omega_n = 1, \hat{y}_n \neq y_n] + 1[\omega_n = 0, y_{dn} \neq y_n] \qquad \textbf{(IP-Main)}
$$

$$
\text{s.t.} \quad |D(z) - D(z')| \leq \varepsilon, \forall z, z' \in \mathcal{Z} \qquad \textbf{(Disparity)}
$$

$$
\frac{\sum_{n=1}^{N} \omega_n \cdot 1[z_n = z]}{\sum_{n=1}^{N} 1[z_n = z]} \geq (1 - \delta_z), \forall z \in \mathcal{Z} \qquad \textbf{(Abstention Rate)}
$$

$$
\sum_{n=1}^{N} 1[\omega_n = 1, \hat{y}_n \neq y_n, z_n = z] + 1[\omega_n = 0, y_{dn} \neq y_n, z_n = z] \leq e'_z \sum_{n=1}^{N} 1[z_n = z], \forall z \in \mathcal{Z}
$$
$$
\textbf{(No Harm)}
$$

$$
\omega_n \in \{0, 1\}, f_n \in \{0, 1\}, \forall n.
$$

Where $y_d$ is the human annotation, $D(z) = \frac{\sum_{n=1}^{N} 1[\hat{y}_n=1, z_n=z, \omega_n=1] + 1[y_{dn}=1, z_n=z, \omega_n=0]}{\sum_{i=1}^{N} 1[z_n=z]}$. The modification does not add any new constraint, the computational complexity remains the same. And Stage II can still be applied directly.

