# Fair Classifiers that Abstain without Harm

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

. For experiments that extend to multiple groups, we direct the reader to Appendix E. Throughout this section, we set $\delta_z = \delta$ across all groups, meaning that each group is constrained by the same upper limit on the permissible rate of abstention. We rigorously evaluated our proposed method, `FAN` against two established baselines: `LTD` (Madras et al., 2018b) and `FSCS` (Lee et al., 2021), as demonstrated in Table 1. For the `LTD` baseline, we employ the learning-to-reject framework, specifically referring to Equation 4 in (Madras et al., 2018b)[3]. We draw upon three real-world datasets for our experiments: `Adult` (Dua & Graff, 2017), `Compas` (Bellamy et al., 2018), and `Law` (Bellamy et al., 2018). During the training phase, we adhere to the Equalized Odds fairness criterion, incorporating two separate constraints. To facilitate a straightforward interpretation of our findings, we compute the average disparity in both the true positive and true negative rates. Due to space constraints, the details of data preprocessing and model setting can be found in Appendix E.

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

**Comparison to Baseline Optimal Classifier.** We conduct a comprehensive set of experiments to examine the impact of `FAN` on both disparity and accuracy. Figure 4 depicts the performance metrics when applying EO on the training and test data for the Adult dataset. These results offer a comparative benchmark against the baseline optimal classifier $h$, facilitating a precise assessment of the degrees to which `FAN` either enhances or compromises model performance in both fairness and accuracy dimensions. Additional results concerning multiple datasets and fairness criteria are provided in Appendix E. The figure elucidates that `FAN` successfully optimizes for a more equitable model without incurring a loss in accuracy. Notably, as the permissible abstention rate ($\delta$) increases, both demographic groups experience a significant improvement in accuracy, while simultaneously reducing the overall disparity. These findings indicate that `FAN` has the ability to train models that are both fairer and more effective, particularly when higher levels of abstention are allowed.

**IP Solution.** Figure 5 is the IP solution corresponding to Figure 4. We observe that each constraint in Problem **IP-Main** is strictly satisfied. As $\varepsilon$ increases, the impact on reducing disparity diminishes due to the inherent allowance for greater disparity in the IP formulation, reducing the need for strict control measures. Intriguingly, as $\delta$ increases, the augmentation in accuracy for both demographic groups remains relatively stable across different configurations. This stability, however, is mainly

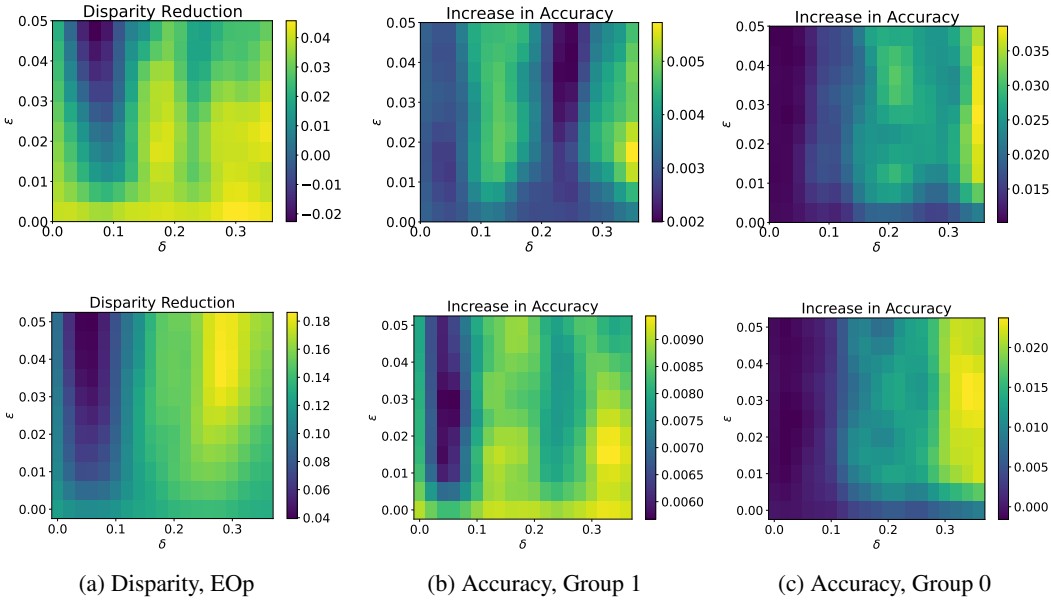

Figure 4: Disparity reduction and increased accuracy for each group performed on `Adult`, compared to baseline optimal classifier. The first row shows the performance on the training data while the second row is on test data. (a) demonstrates the disparity reduction in terms of Equal Opportunity, while (b) and (c) showcase the increases in accuracy for group 1 and 0, separately. x-axis represents the maximum permissible abstention rate, while y-axis represents the maximum allowable disparity.

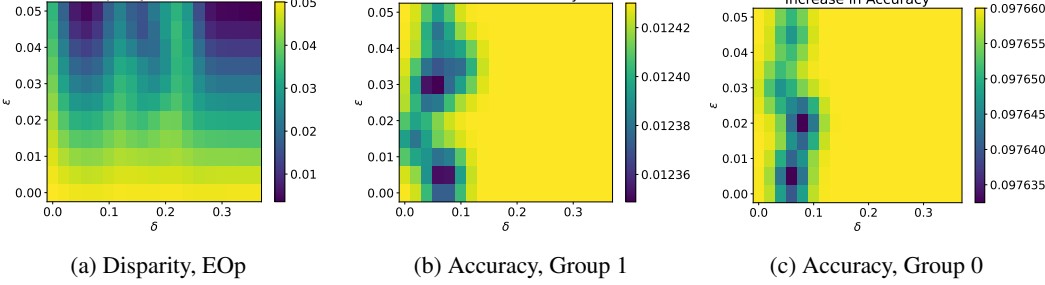

(a) Disparity, EOp      (b) Accuracy, Group 1      (c) Accuracy, Group 0

Figure 5: This figure illustrates the result of the IP solution, under the same setting of Figure 4.

because the accuracy is already approaching an optimal level in this specific experimental setup, leaving minimal scope for substantial further improvements. A side-by-side comparison between Figure 5 and Figure 4 reveals a strong alignment between the results derived from the training data and those obtained from the IP formulation. This concordance

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

$, problem 7 is feasible under Equal Opportunity iff for all $z \in \mathcal{Z}$,*

$$(1)\ \delta_z \leq 1 - \tau_z\ ;\ (2)\ \delta_z \geq e_z - \tau_z\ ;\ (3)\ \delta_z \geq \frac{\tau_z - \eta_z e_z - 1}{1 - (1 + \eta_z)e_z}\ ;\ (4)\ \delta_z \geq \frac{-\eta_z e_z}{2 - (1 + \eta_z)e_z} \tag{8}$$

**Example C.2.** $e_z = 0.3, \eta_z = 0$, if $\tau_z = 0.6, \delta_z \leq 0.4$; if $\tau_z = 0.2, 0.1 \leq \delta_z \leq 0.8$.

An intriguing observation is that under Equal Opportunity, $\varepsilon$ is not a factor that affects feasibility. This implies that IP can achieve exact fairness, eliminating any form of disparity.

# D  PROOF

## D.1  NOTATIONS

For simplicity, we introduce some notations that will be used in the proof. Figure 8 illustrates the distribution ratios of different regions of two groups.

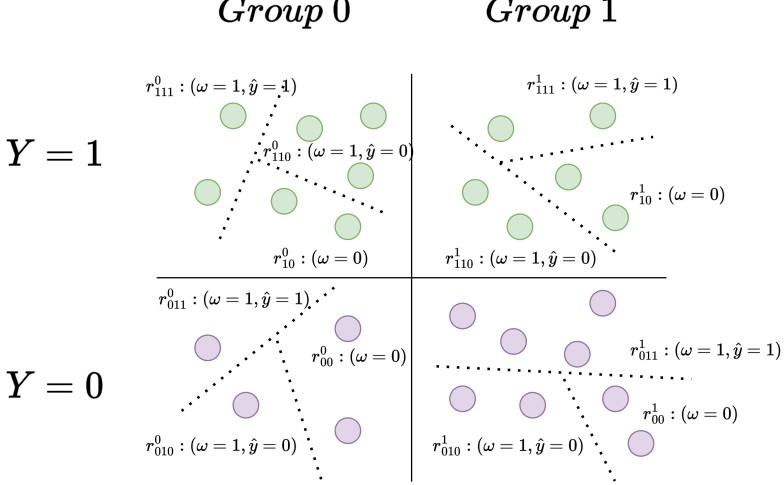

Figure 8: Division of individuals into four regions based on their sensitive attribute and label. The ratio, denoted as $r^z ijk$, represents the proportion of individuals of group z with specific combinations of $y = i, \omega = j$, and $\hat{y} = k$. For $\omega = 0$, define $r_{i0}^z = r_{i00}^z + r_{i01}^z$. By definition, $r_{111}^z + r_{110}^z + r_{10}^z + r_{011}^z + r_{010}^z + r_{00}^z = 1$ holds true, ensuring the sum of all ratios within group z is equal to one. Define $r^z = \{r_{111}^z, r_{110}^z, r_{10}^z, r_{011}^z, r_{010}^z, r_{00}^z\}$.

Note that $e_z' = (1 + \eta_z)e_z$. For notation simplicity, define $a_z' = 1 - e_z'$.

## D.2  PROOF OF LEMMA D.1

**Lemma D.1.** *Under fairness notion Demographic Parity (Equal Opportunity), if Problem **IP-Main** is feasible, there must exist a solution such that the resulting classifier does not abstain any samples with label 0, i.e., for all n such that $y_n = 0$, it holds that $\omega_n = 1$.*

Lemma D.1 indicates abstaining data samples with label 0 are not required by IP.

*Proof.* Without loss of generality, we consider group 0. Since Problem **IP-Main** is feasible, there exist a $r^0$ that Constraints **Abstention Rate, Disparity, No Harm** hold.

Constraint **No Harm** can be written as

$$\frac{r^0_{110} + r^0_{011}}{r^0_{111} + r^0_{110} + r^0_{011} + r^0_{010}} \leq e'_0 \Leftrightarrow a'_0(r^0_{110} + r^0_{011}) \leq e'_0(r^0_{111} + r^0_{010}) \tag{9}$$

Constraint **Abstention Rate** can be written as

$$r^0_{10} + r^0_{00} \leq \delta_0 \tag{10}$$

For Demographic Parity, constraint **Disparity** indicates

$$r^z_{111} - r^z_{010} - \tau_z - \varepsilon \leq r^0_{111} - r^0_{010} - \tau_0 \leq r^z_{111} - r^z_{010} - \tau_z + \varepsilon, \forall z \tag{11}$$

If the feasible solution $r^0_{00} > 0$, define $r^{0\prime}_{010} = r^0_{010} + r^0_{00}, r^{0\prime}_{00} = 0$, then it's not hard to verify that constraints 9, 10 and 11 still hold for $r^{0\prime}_{010}$ and $r^{0\prime}_{00} = 0$. $r^{0\prime}_{00} = 0$ indicates abstaining no individual with label 0.

For Equal Opportunity, constraint **Disparity** indicates

$$\frac{r^z_{111}}{\tau_z} - \varepsilon \leq \frac{r^0_{111}}{\tau_0} \leq \frac{r^z_{111}}{\tau_z} + \varepsilon, \forall z \tag{12}$$

Similarly, if the feasible solution $r^0_{00} > 0$, define $r^{0\prime}_{010} = r^0_{010} + r^0_{00}, r^{0\prime}_{00} = 0$, then it's not hard to verify that constraints 9, 10 and 12 still hold for $r^{0\prime}_{010}$ and $r^{0\prime}_{00} = 0$. $r^{0\prime}_{00} = 0$ indicates abstaining no individual with label 0.

## D.3 PROOF OF THEOREM 4.1

By Lemma D.1, we have Problem **IP-Main** is feasible iff there exist a solution that $r^z_{00} = 0, \forall z$.

For any two group $z = 0, 1$, without loss of generality, assume $\tau_1 \leq \tau_0$. Define $\Delta\tau = \tau_0 - \tau_1$.

Similar to 10, 9 and 11, the constraint of **IP-Main** can be written as

$$\begin{cases} r^0_{111} - r^0_{010} - \Delta\tau - \varepsilon \leq r^1_{111} - r^1_{010} \\ r^0_{111} - r^0_{010} - \Delta\tau + \varepsilon \geq r^1_{111} - r^1_{010} \\ r^1_{10} \leq \delta_1 \\ r^0_{10} \leq \delta_0 \\ a'_0(r^0_{110} + r^0_{011}) \leq e'_0(r^0_{111} + r^0_{010}) \\ a'_1(r^1_{110} + r^1_{011}) \leq e'_1(r^1_{111} + r^1_{010}) , \\ r^1_{111} + r^1_{10} + r^1_{110} = \tau_1 \\ r^0_{111} + r^0_{10} + r^0_{110} = \tau_0 \\ r^1_{010} + r^1_{011} = 1 - \tau_1 \\ r^0_{010} + r^0_{011} = 1 - \tau_0 \\ r^1, r^0 \geq \mathbf{0} \end{cases} \tag{13}$$

Let $r^1_{011} = 1 - \tau_1 - r^1_{010}, r^0_{011} = 1 - \tau_0 - r^0_{010}, r^1_{110} = \tau_1 - r^1_{111} - r^1_{10}, r^0_{110} = \tau_0 - r^0_{111} - r^0_{10}$, 13 becomes

$$\begin{cases} r_{111}^0 - r_{010}^0 - \Delta\tau - \varepsilon \leq r_{111}^1 - r_{010}^1 \\ r_{111}^0 - r_{010}^0 - \Delta\tau + \varepsilon \geq r_{111}^1 - r_{010}^1 \\ r_{10}^1 \leq \delta_1 \\ r_{10}^0 \leq \delta_0 \\ r_{111}^1 + a_1 r_{10}^1 + r_{010}^1 \geq a_1' \\ r_{111}^0 + a_m r_{10}^0 + r_{010}^0 \geq a_0' \;, \\

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

$$
(1)\ \delta_1 \le 1 - \tau_1 \ ;\ (2)\ \delta_1 \ge e_1 - \tau_1 \ ;\ (3)\ \delta_1 \ge \frac{\tau_1 - \eta_1 e_1 - 1}{1 - (1 + \eta_1) e_1} \ ;\ (4)\ \delta_1 \ge \frac{-\eta_1 e_1}{2 - (1 + \eta_1) e_1}
\tag{29}
$$

### D.7 Proof of Theorem 4.5

Note that when the equal abstention rate constraints are added, Lemma D.1 still holds. The reason is $r_{00}^1 = r_{00}^0 = 0$ already yields equal abstention rate, since the abstention rates of individuals with negative label are both 0.

Similar to 13, the constraints of Problem 3 can be written as

$$
\begin{cases}
r_{111}^0 - r_{010}^0 - \Delta\tau - \varepsilon \leq r_{111}^1 - r_{010}^1 \\
r_{111}^0 - r_{010}^0 - \Delta\tau + \varepsilon \geq r_{111}^1 - r_{010}^1 \\
r_{10}^1 \leq \delta_1 \\
r_{10}^0 \leq \delta_0 \\
r_{10}^1 \leq r_{10}^0 \frac{\tau_1}{\tau_0} + \tau_1 \sigma_1 \\
r_{10}^1 \geq r_{10}^0 \frac{\tau_1}{\tau_0} - \tau_1 \sigma_1 \\
a_0'(r_{110}^0 + r_{011}^0) \leq e_0'(r_{111}^0 + r_{010}^0) \\
a_1'(r_{110}^1 + r_{011}^1) \leq e_1'(r_{111}^1 + r_{010}^1) \\
r_{111}^1 + r_{10}^1 + r_{110}^1 = \tau_1 \\
r_{111}^0 + r_{10}^0 + r_{110}^0 = \tau_0 \\
r_{010}^1 + r_{011}^1 = 1 - \tau_1 \\
r_{010}^0 + r_{011}^0 = 1 - \tau_0 \\
r^1, r^0 \geq \mathbf{0}
\end{cases}
\tag{30}
$$

Compare to 13, the additional constraints of $r_{10}^1$ are $r_{10}^0 \frac{\tau_1}{\tau_0} - \tau_1 \sigma_1 \leq r_{10}^1 \leq r_{10}^0 \frac{\tau_1}{\tau_0} + \tau_1 \sigma_1$. Plug in them and extract the condition of $r_{10}^1$ yields

$$
\begin{cases}
0 \leq r_{10}^1 \leq \delta_1 \\
r_{10}^1 \geq \dfrac{a_1' - r_{010}^1 - r_{111}^1}{a_1'} \\
r_{10}^1 \leq \tau_1 - r_{111}^1 \\
\tau_0 r_{111}^1 + \tau_1 r_{10}^0 \leq \tau_0 \tau_1 (1 + \sigma_1) \\
\tau_0 (a_1' - r_{010}^1 - r_{111}^1) \leq a_1' r_{10}^0 + \tau_1 \sigma_1 a_1'

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

First, we evaluate `FAN` on a Multi Group scenario using the `Law`. The sensitive attribute `race` includes categories `White`, `Black`, `Asian`, `Hispanic`, and `Other`. Figure 19 illustrates the minimum disparity reduction and minimum increase in accuracy across all groups, i.e., the group with the worst performance.

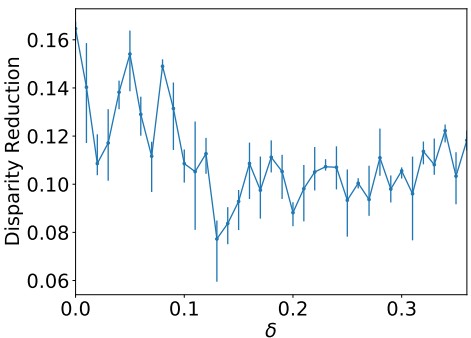 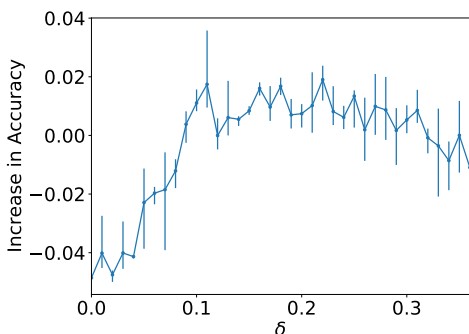

Figure 19: Performance of `FAN` on Multi Group Scenario: Evaluating the effectiveness on the `Law` under Demographic Parity. Left: Minimum disparity reduction across all groups. Right: Minimum increase in accuracy across all groups. `FAN` consistently reduces disparity, while achieving higher accuracy when abstention rate is not too low.

We also evaluate `FAN` on a Multi Group scenario using the `Compas`. The sensitive attribute `race` includes categories `Caucasian`, `African-American`, `Asian`, `Hispanic`. Figure 20 illustrates the minimum disparity reduction and minimum increase in accuracy across all groups, i.e., the group with the worst performance.

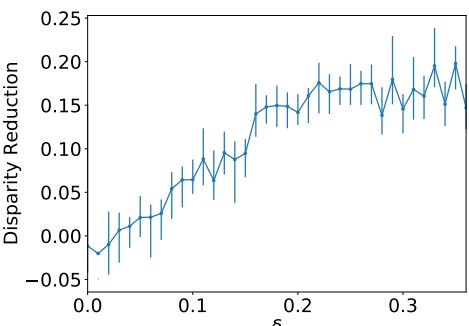 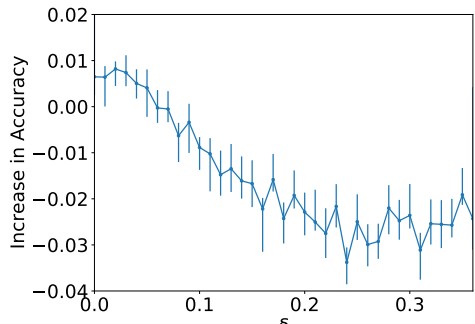

Figure 20: Performance of `FAN` on Multi Group Scenario: Evaluating the effectiveness on the `Compas` under Equal Opportunity. Left: Minimum disparity reduction across all groups. Right: Minimum increase in accuracy across all groups. `FAN` consistently reduces disparity, while achieving higher accuracy when abstention rate is not too low.

We also evaluate `FAN` on a Multi Group scenario using the `Adult`. The sensitive attribute `race` includes categories `White`, `Black`, `Asian-Pac-Islander`. Figure 21 illustrates the minimum disparity reduction and minimum increase in accuracy across all groups, i.e., the group with the worst performance.

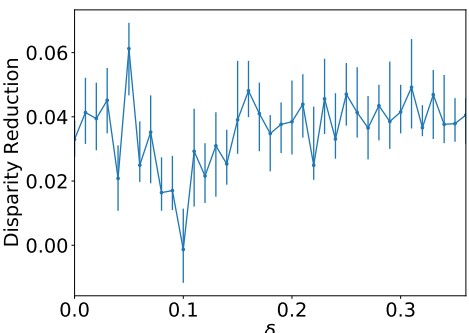 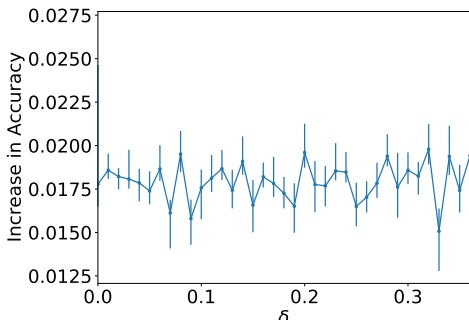

Figure 21: Performance of `FAN` on Multi Group Scenario: Evaluating the effectiveness on the `Adult` under Equal Opportunity. Left: Minimum disparity reduction across all groups. Right: Minimum increase in accuracy across all groups. `FAN` consistently reduces disparity, while achieving higher accuracy when abstention rate is not too low.

### E.6 PREDICTION CONSISTENCY