# OpenReview forum: "Fair Classifiers that Abstain without Harm"
_ICLR.cc/2024/Conference — ICLR 2024 poster_

### Official Review · Reviewer_7v7y · 2023-11-01

**Soundness:** 3 good
**Presentation:** 3 good
**Contribution:** 2 fair
**Rating:** 6
**Confidence:** 3

**Summary:**

The authors shed light on the challenge of fair classification with the option of abstaining from a prediction. Their objective is to develop a post-hoc fair classification algorithm that meets the following four criteria: 1) the accuracy across groups should be almost as good as the provided classifier, 2) the fairness criteria should be maintained within a prescribed threshold, 3) the feasibility of achieving both the aforementioned accuracy and fairness, given a specified abstention rate, must be ascertainable, and 4) a range of fairness criteria can be applied. The suggested algorithm employs a mixed integer problem solver to determine the best abstentions and label flips that minimize classification errors while simultaneously ensuring the desired levels of accuracy, fairness, and abstention rate. Subsequently, the algorithm constructs a neural network to predict these optimal abstentions and label flips. As theoretical findings, the authors highlight the essential conditions needed for the accuracy, fairness, and abstention rate constraints to be feasible, considering various fairness criteria, including demographic parity, equalized odds, and equal opportunity. Experimental outcomes indicate that the new algorithm maintains fairness without compromising accuracy, a distinction from existing methods, which often trade off one for the other.

**Strengths:**

1. The paper is clearly written and easy to understand.

2. The experimental results robustly confirm the improvement of fairness without compromising accuracy, distinguishing the proposed algorithm from existing methods like LTD and FSCS.

3. The theoretical analyses provide insightful results that may elucidate the conditions under which the best classifier with optimal abstention satisfies the requirements. This could potentially characterize the Bayes optimal classifier with abstentions, contributing valuable insights into the trade-off between accuracy and fairness.

**Weaknesses:**

1. The method proposed evalates the fairness of the learning classifier using abstained sample. This approach seems impractical in real-world scenarios, where actionable decisions are often needed even for abstained cases. Previous studies, like those of LTD and FSCS, suppose that abstained decisions default to human intervention. Consequently, the fairness of the complete system should encompass both algorithmic and human decisions. Unlike these studies, the proposed algorithm's rationale behind its fairness constraints remains ambiguous. It would benefit readers if the authors presented a real-world scenario validating their algorithm's constraints.

2. Using both an error rate objective function and a no-harm constraint seems redundant as they essentially serve the same purpose.

3. The authors state that their algorithm upholds hard fairness constraints. However, the optimization problem they designed utilizes an approximate fairness constraint. Moreover, the second stage might infringe upon this strict fairness constraint since it merely constructs a function that mimics the labels derived from Stage I.

4. While the fairness requirements of the proposed algorithm differ from those in existing studies (LTD and FSCS), the authors use the evaluation metric of the proposed algorithm's fairness in the experiments. This approach is unfair to the existing methods.

**Questions:**

1. Can the authors illustrate a specific scenario in which the constraints of their proposed algorithm are essential?

---

> ### Author Response · Authors · 2023-11-16
>
> We thank the reviewer for their positive feedback and constructive comments.
>
> > The method proposed evalates the fairness of the learning classifier using abstained sample. This approach seems impractical in real-world scenarios, where actionable decisions are often needed even for abstained cases. Previous studies, like those of LTD and FSCS, suppose that abstained decisions default to human intervention. Consequently, the fairness of the complete system should encompass both algorithmic and human decisions. Unlike these studies, the proposed algorithm's rationale behind its fairness constraints remains ambiguous. It would benefit readers if the authors presented a real-world scenario validating their algorithm's constraints.
>
> **Human annotations on abstained decision.** Our motivation is indeed that the abstained data samples will have to go through human annotations. In this case, bounding the abstention rate control can help the practitioner reduce cost and delay of human annotation. Also, low abstention rate likely leads to high quality of human annotation, under the same resource given (especially when we operate in a low-recourse and real-time application).
>
> Our work focuses on making the abstention process fair and in a more controlled fashion **before** we interact with the human annotators. Note that FSCS does not consider human intervention as part of model training as well. While LTD incorporates human annotation in model training (Eqn. 3 in their paper), we claim this may be too ideal and not realistic in most applications. In real world applications, the training data size is usually very large, and it’s rare that the entire dataset has corresponding human annotation - This also motivates us to impose abstention rate control to reduce resource burden.
>
> Even in the case that human annotation exists for training data, our method can be applied with some modifications of the IP. For example, under Demographic Parity,
>
> (We also add this to Appendix F in the updated version of our paper.)
>
> $\\min_{\\omega, f}  \\sum_{n=1}^{N} 1[\\omega_n = 1, \\hat y_n \\neq y_n] + 1[\\omega_n = 0, y_{dn} \\neq y_n]$
>
> $s.t.     | \\frac{\\sum_{n = 1}^{N} 1[\\hat y_n = 1, z_n = z, \\omega_n=1] + 1[y_{dn} = 1, z_n = z, \\omega_n=0]}{\\sum_{i=1}^{N} 1[z_n = z]} - \\frac{\\sum_{n = 1}^{N} 1[\\hat y_n = 1, z_n = z', \\omega_n=1] + 1[y_{dn} = 1, z_n = z', \\omega_n=0] }{\\sum_{i=1}^{N} 1[z_n = z']} | \\leq \\varepsilon, \\forall z, z' \\in \\mathcal Z$
>
> $\qquad$ (Same abstention rate constraint)
>
> $\qquad \\sum_{n=1}^{N} 1[\\omega_n = 1, \\hat y_n \\neq y_n, z_n=z] + 1[\\omega_n = 0, y_{dn} \\neq y_n, z_n=z] \\leq (1 + \eta_z) e_z \\sum_{n=1}^{N} 1[z_n=z], \\forall z \\in \\mathcal Z$
>
>
>
>
> Where $y_{d}$ is the human annotation. The modification does not add any new constraint, the computational complexity remains the same. And Stage II can still be applied directly.
>
> > Can the authors illustrate a specific scenario in which the constraints of their proposed algorithm are essential?
>
> The fairness constraint: Fairness holds significant importance in various contexts. For example, In lending, it’s against bias, guaranteeing equitable financial access across diverse races, genders, and more. Similarly, in facial recognition, it serves to prevent discriminatory misidentifications.
>
> The abstention rate constraint: It is essential in scenarios that resource is limited and only a small piece of data can be left to human annotators. It is also important to bound the abstention rate when a near real-time decision will have to be made.
>
> The No harm constraint: the no harm constraint ensures that no group becomes worse-off behind the abstaining process. In the answer below we illustrate a scenario that shows the importance of no harm.

---

> ### Author Response · Authors · 2023-11-16
>
> > Using both an error rate objective function and a no-harm constraint seems redundant as they essentially serve the same purpose.
>
> Improving overall error rate doesn’t imply no harm on group level accuracy. In the example below (two groups with the same size), the overall error rate decreases from 1st row to 2nd, but Group2 is compromised with an increasing error rate.
>
> | Group1 Error Rate | Group2 Error Rate | Overall Error Rate |
> |-------------------|-------------------|--------------------|
> | 0.2               | 0.1               | 0.15               |
> | 0.1               | 0.15              | 0.125              |
>
> On the contrary, only imposing no harm constraint also does not lead to an overall optimal solution as well. The no harm constraint provides a feasible region (with many solutions falling inside), while the objective function further chooses the one inside the region that minimizes overall error rate.
>
> > The authors state that their algorithm upholds hard fairness constraints. However, the optimization problem they designed utilizes an approximate fairness constraint. Moreover, the second stage might infringe upon this strict fairness constraint since it merely constructs a function that mimics the labels derived from Stage I.
>
> **Regarding fairness constraint in Stage I.** Strict fairness is achieved directly by setting $\varepsilon=0$. We impose the soft fairness constraint because it’s commonly used in the literature and a more general version, and, in our case, leads to better abstention policies.
>
> **Regarding Stage II.** Our theoretical results help us understand the fundamental limit in that aspect but admit that this is probably only approximately true for the surrogate models. As discussed in Section 5 (Stage II Analysis), \texttt{FAN} may not strictly meet the hard constraint due to the surrogate model training of \texttt{AB} and \texttt{FB}. This issue widely exists in the generalization of surrogate models and is not unique to our method. In the deep learning domain, theoretically bounding the performance gap between predictions and ground truth is generally challenging. Thus, to assess the effectiveness of surrogate models, we conduct experiments to demonstrate their accuracy. As depicted in Table 2, the surrogate model generally performs well in learning IP solutions. For example, when $\delta=0.2$ for \texttt{Adult}, AB achieves 92.20%, 89.93%, 88.48% accuracy under DP, EO, EOd, respectively, meanwhile FB achieves 97.79%, 95.33%, 95.86%.
>
> > While the fairness requirements of the proposed algorithm differ from those in existing studies (LTD and FSCS), the authors use the evaluation metric of the proposed algorithm's fairness in the experiments. This approach is unfair to the existing methods.
>
> The evaluation of fairness is consistent with the baselines. The fairness measure of FSCS is naturally the same as ours as they don't have human annotator in training as well. Please kindly let us know if there is a specific mismatch the reviewer sees.
>
> For LTD, we consider specifically their ``Learning to reject’’ method, which does not incorporate human intervention in training either, as the datasets we adopted don’t have human annotation.

---

> ### Author Response · Authors · 2023-11-22
> **Rebuttal period ends soon**
>
> We express our gratitude once more to the reviewer for their comments. As the deadline for the rebuttal period is approaching, we kindly inquire if the reviewer has any additional comments or questions. We are happy to address any further inquiries.

---

### Official Review · Reviewer_RYdg · 2023-11-09

**Soundness:** 3 good
**Presentation:** 4 excellent
**Contribution:** 3 good
**Rating:** 6
**Confidence:** 4

**Summary:**

The paper addresses the problem of training fair classifiers, but also caters to other constraints such as not reducing group-wise accuracy and providing a "do not predict" option. The main idea of the paper is to increase the feasibility region of the fair classification problem (and other constraints) by abstaining and flipping predictions. The overall constrained problem is solved by Integer Programming. To use the models on unseen data, the paper trains surrogate models to the Integer Programming solution.

**Strengths:**

1. The paper is quite well-written. Most design choices are appropriately motivated. Terminology is clean and easy to understand despite the large number of components involved.

2. The experimental results are encouraging.

3. The paper solves a mix of problems that are all quite useful: fairness, abstaining from making decisions, and not reducing accuracy for groups in the data. All of these components are individually addressed elsewhere in prior work, but putting them all together is a nice contribution.

**Weaknesses:**

I think the paper needs to address a couple of points before it is ready for publication:

1. The paper claims to provide hard constraint satisfaction guarantees but does not discuss how these guarantees are supposed to hold when replacing AB and FB modules with surrogate models, and when replacing the true label predictor with a surrogate model. Does the generalization ability of these surrogate models not affect the constraint satisfaction? If yes, how? Or is that the guarantees only hold when assuming Bayers Optimal predictors?

2. On a related note, the paper should provide some discussion into the functional form of the surrogate models. In the appendix, the paper mentions using different Neural Net architectures for different datasets. Is there some guidance on how the architectures should be selected? Should one select the optimal architectures using hyperparameter tuning in isolation (one surrogate model at a time) or should the tuning procedure consider the whole end-to-end Integer Program?

3. Perhaps I missed it, but the paper does not provide information about training cost (e.g., wallclock time). Seeing how the training cost scales with number of data points is essential in judging the effectiveness of the proposed procedure.

**Questions:**

1. It would be great to get the answers to points 1-3 in the "Weaknesses" section.

2. Is it possible to extend no-harm to individual samples, e.g., the prediction on individual samples should not flip from positive to negative?

---

> ### Author Response · Authors · 2023-11-16
>
> We thank the reviewer for their positive feedback and constructive comments.
>
> > The paper claims to provide hard constraint satisfaction guarantees but does not discuss how these guarantees are supposed to hold when replacing AB and FB modules with surrogate models, and when replacing the true label predictor with a surrogate model. Does the generalization ability of these surrogate models not affect the constraint satisfaction? If yes, how? Or is that the guarantees only hold when assuming Bayes Optimal predictors?
>
> The baseline does not need to a Byes Optimal predictor - indeed it can be any classifier. Our guarantees, imposed by the hard constraints, will hold when we consider a sufficiently large hypothesis space that encodes all possible combinations of classification outcomes for the training data.
>
> Practically, as discussed in Section 5, Stage II Analysis, FAN may not strictly meet the hard constraint due to the surrogate model training of AB and FB. This issue widely exists of the generalization in the surrogate model and is not unique to our method; in the deep learning domain, theoretically bounding the performance gap between predictions and ground truth is generally challenging. This extension is a bit derivative in our opinion and didn’t want to distract the readers but we will add discussions in our next version. In the current draft, to assess the effectiveness of surrogate models, we conduct experiments to demonstrate their accuracy. As depicted in Table 2, the surrogate model generally performs well in learning IP solutions. For example, when $\delta=0.2$ for Adult, AB achieves 92.20%, 89.93%, 88.48% accuracy under DP, EO, EOd, respectively, meanwhile FB achieves 97.79%, 95.33%, 95.86%.
>
> > On a related note, the paper should provide some discussion into the functional form of the surrogate models. In the appendix, the paper mentions using different Neural Net architectures for different datasets. Is there some guidance on how the architectures should be selected? Should one select the optimal architectures using hyperparameter tuning in isolation (one surrogate model at a time) or should the tuning procedure consider the whole end-to-end Integer Program?
>
> The surrogate model should be tuned independently. After obtaining the IP results ($\omega, f$), the training of the surrogate model itself is a standard classification task. There are many existing works on hyperparameter tuning (learning rate, regularization, etc.). We can also split the data into train/validation sets and use the validation data to select the proper architecture.
>
> The IP itself doesn’t incorporate any tunable parameters. The abstention rate bound $\delta$, disparity bound $\varepsilon$, and allowable accuracy reduction $\eta$ are all fixed parameters and should be specified by the policymaker at the beginning. In Section 4, we have revised the reference of them to "design parameters" instead of "hyperparameters" to avoid confusion accordingly.
>
> > Perhaps I missed it, but the paper does not provide information about training cost (e.g., wallclock time). Seeing how the training cost scales with the number of data points is essential in judging the effectiveness of the proposed procedure.
>
> We thank the reviewer for pointing this out and report average time cost in the updated version, in Appendix E.9. For example, when using MLP (details in E:Neural Network), for Adult under DP, the IP solution takes 36.76s (1.08), training AB needs 14.33s (1.02), training FB needs 14.72s (1.65). The numbers in parentheses are std.
>
> > Is it possible to extend no-harm to individual samples, e.g., the prediction on individual samples should not flip from positive to negative?
>
> As we have a general IP formulation, it is possible to impose the additional constraint to avoid flipping from positive to negative in Stage I:
>
> $$
> \sum_{i=1}^{N} 1[\hat{y}_{bn} = 1, f_n = 1] = 0
> $$
>
> while Stage II remains the same. Note that adding new constraints will also reduce the feasibility region of the IP, which leads to interesting future work. In our current approach, although we do not impose this hard constraint, we prioritize flipping individuals with lower confidence scores.

---

> > ### Author Response · Authors · 2023-11-23
> > **Rebuttal period ends today**
> >
> > We express our gratitude once more to the reviewer for their comments. As the deadline for the rebuttal period is approaching, we kindly inquire if the reviewer has any additional comments or questions. We are happy to address any further inquiries.

---

### Official Review · Reviewer_GsZB · 2023-11-10

**Soundness:** 3 good
**Presentation:** 3 good
**Contribution:** 3 good
**Rating:** 8
**Confidence:** 4

**Summary:**

The paper focuses on the problem of selective binary classification under fairness, abstention and harm constraints.  To accommodate for all the constraints the authors propose a framework with two basic components: a mechanism deciding from which instances the classifier should abstain and a mechanism deciding to flip the predictions of the classifier. Using these components the authors first formulate the problem of minimizing the classification error under disparity, abstention rate, and no harm constraints with IP.  Then they propose solving the IP problem for a small dataset and use these solutions to train models on the abstention and flipping component to predict near optimal decisions on unseen data. Finally, the authors deploy their framework on several datasets and compare it with competitive baselines.

**Strengths:**

The work appears to be the first to consider the problem of selective classification under fairness, abstention rate, and no harm constraints on the same time. The proposed approach seems quite interesting especially for being flexible about the type of fairness constraints that one may impose.  In addition,  achieving fairness guarantees without sacrificing accuracy seems of great importance for real world applications.

The paper appears very-well structured and nicely written. The authors clearly describe their contributions and sufficiently discuss relation to contemporary literature. Moreover, they present the experimental setup in detail. The experimental evaluation appears thorough and the results seem promising.

**Weaknesses:**

Even though the paper is nicely and clearly written, there are a few points that could confuse the reader:

In the Paragraph “Stage I: Integer Programming. We approximate h_A and h_F…” “approximate” is confusing as  $h_A$ and $h_F$  are already defined as binary parameters.

In the optimization problem in section 3.1 the abstention rate and the no harm constraints are not defined for any $z \in \mathcal{Z}$, whereas in the IP-Main these constraints are defined for each $z  \in \mathcal{Z}$. If IP-Main is a way to practically solve of the optimization problem in section 3.1, the definitions should be consistent. If there is a reason why these definitions should be different, this reason should be made clear.

It is not clear what is the motivation for section 4.2. Since in IP-Main one does provide a desired abstention rate constraint for each $z \in \mathcal{Z}$ it is not clear what benefit would bring further constraints on the difference on the abstention rates. Especially in the case that the cardinality of $\mathcal{Z}$ is large, additional pairwise  constraints  for each pair $z,z’ \in \mathcal{Z}$ would add significant overhead in solving (3). Also, in (3) $z’$ is not defined.

The reported results in Figure 3 are over only 5 different runs. One could argue that this is a quite limited evaluation. Given that the results do look promising and the error bars are relatively small, showing results over more runs would strengthen the significance of the results. If there are computational limitations that prevented the authors from evaluating their method for more runs, these should be made clear. The same applies for the results of Table 2. In addition, Table 2 is missing confidence intervals and the type of the error bars in Figure 3 are not  specified.

The authors should consider adding a (brief) discussion on limitations of their approach and on perspectives for future work.



 Typos/Misc:
- 1st paragraph in section 2 “i.e.” —> “i.e.,” and “e.g.” —> “e.g.,”
- 2nd paragraph  “to determine which samples to abstain” is not very clear. Suggestion “to determine from which samples the classifier should abstain”
- Section 4 first paragraph “hyperparameter” —> “hyperparameters”
- Bottom of page 5 “as the models are neural network” —> “as the models are neural networks”
- Top of page 6 right most column of the Table “TBD in 3.1” do the authors mean “TBD in 4.1”?
- “An objection may arise that the model’s excessive abstention from a particular group, while not observed in others.” This seems as an incomplete sentence. What do the authors mean here?
- Missing “.” In footnote 2.
- Page 8 top “DO”—> “DP”
- Conclusion “our abstaining process incur” —> “our abstaining process incurs”

**Questions:**

1. With FB one could use any arbitrary vector of random predictions (not necessarily from a classifier) and learn when to flip then or not. If so why would one need a classifier in the first place?
2. It is not clear what is the motivation for section 4.2. Since in IP-Main one does provide a desired abstention constraint for each $z \in \mathcal{Z}$ why would one would like to further constraint the difference of abstention rate?
3. why would one would like to further constraint the difference of abstention rate? Also it is not clear in what sense “the performance will become worse”. It might more helpful to clarify if the authors mean that the IP problem will be harder to solve or if the solution of the problem will have a higher error rate. Not sure that 4.2 adds much, maybe remove it?

---

> ### Author Response · Authors · 2023-11-16
>
> We thank the reviewer for their encouraging feedback and constructive comments; these help us improve our paper.
>
> > In the Paragraph “Stage I: Integer Programming. We approximate h_A and h_F…” “approximate” is confusing as h_A and h_F are already defined as binary parameters.
>
> Thank the reviewer for pointing this out. We have revised the paragraph as follows:
>
> Stage I: Integer Programming. For a dataset with $N$ individuals, the goal of Stage I is to learn two N-length vectors:
>
> $\\omega = \\{\omega_n\\}_N$ and $f = \\{f_n\\}_N$,
>
> which represent predictions of $h_A(X, h(X))$ and $h_F(X, h(X))$ on this dataset, respectively. In other words, $\\omega_n = h_{A}(x_n, h(x_n)) \\in \\{0, 1\\}$, and $f_n = h_{F}(x_n, h(x_n)) \\in \\{0, 1\\}$.
>
> > It is not clear what is the motivation for section 4.2. Since in IP-Main one does provide a desired abstention rate constraint for each $z \in \mathcal{Z}$. it is not clear what benefit would bring further constraints on the difference on the abstention rates. Especially in the case that the cardinality of $\mathcal{Z}$ is large, additional pairwise constraints for each pair $z, z’ \in \mathcal{Z}$ would add significant overhead in solving (3). Also, in (3) $z’$ is not defined. Also it is not clear in what sense “the performance will become worse”. It might be more helpful to clarify if the authors mean that the IP problem will be harder to solve or if the solution of the problem will have a higher error rate. Not sure that 4.2 adds much, maybe remove it?
>
> The equal abstention rate (as shown in Section 4.2) represents a stricter control added to the system. It is not required in all applications (which is why we do not include it in the IP-Main from the beginning), but it may be necessary in some applications. Below, we explain the details. Some of the discussions are included in the updated paper.
>
> **Motivation of equal abstention rate.**
>
> We address the motivation from two perspectives.
>
> 1) In IP-Main, the abstention rate is controlled for each group independently. However, constraining the abstention rate doesn’t imply the rates would be equalized. For example, there are two groups with an abstention rate constraint being 0.1. But they can end up with one being 0.01, and another being 0.1, creating a gap in the abstentions. It could be deemed as "unfair" in some instances. Imposing the constraint in Section 4.2 serves the purpose of equalizing the abstention rate among groups.
>
> 2) Moreover, the additional constraint also equalizes the abstention rate separately for the 'qualified' (y=1) and 'unqualified' (y=0) individuals. This division proves beneficial in systems where differing attention and focus are directed towards y=1 or y=0. For example, in disease prediction, those who truly have the disease (y=1) are the major focus of the system, and the corresponding equal abstention rate may be considered more important than for those without the disease.
>
> **Computational complexity.**
>
> The reviewer understands correctly that equal abstention rate constraint increases computational complexity of the optimization problem. There’s no free lunch; if the 'fair abstention rate' is favored, then some sacrifice in time complexity is needed. The choice is left to the decision maker/policy designer to make.
>
> It’s also worth mentioning that solving IP problems and training surrogate models are both offline processes, only required once before the deployment of FAN. The inference overhead is not impacted and still small.
>
> In addition, it’s possible to significantly reduce such sacrifice from $O(N^2)$ to $O(N)$. In an N group scenario, imposing equal abstention rates adds $2 N^2$ more constraints to the system. (2 comes from separate control of y=1 and y=0.) We now show that it can be easily reduced to $2 N$ practically if minor approximation is allowed. Denote $r_{z, 1}$ as the abstention rate of qualified individuals (y=1) in group z. Instead of requiring $| r_{z, 1} - r_{z’, 1} | \leq \sigma_{1}, \forall z, z’$, we only require $| r_{z, 1} - r_{0, 1} | \leq \frac{1}{2} \sigma_{1}, \forall z$, i.e., only bound the difference of all groups from group 0. It’s slightly more strict but reduces much time cost.
>
> **Explanation of why the performance became worse.**
>
> ``Worse performance’’ represents a higher overall error rate (the objective). The reason is that the adding of new constraints reduces the feasible region, while the objective remains the same. It's a fairness-accuracy trade-off.

---

> ### Author Response · Authors · 2023-11-16
>
> > The reported results in Figure 3 are over only 5 different runs … computational limitations… The same applies for the results of Table 2. In addition, Table 2 is missing confidence intervals and the type of the error bars in Figure 3 are not specified.
>
> We're conducting new experiments to increase the number of runs. We will be able to present partial results before the rebuttal period ends and will revisit this question. Additionally, we will ensure to extend more runs to all experiments in the future version. In the current version, the error bar is small showing a convincing improvement of our method, while we acknowledge that more runs would be preferable.
>
> > The authors should consider adding a (brief) discussion on limitations of their approach and on perspectives for future work.
>
> We add the following to Conclusion & Discussion:
>
> Interesting future directions for our research involve extending our method beyond binary classification tasks to encompass multi-class scenarios. Preliminary considerations suggest transforming the problem into a series of binary classification tasks, with additional design required to refine the flipping mechanism. Another avenue is to include a human subject study, incorporating human annotation for abstained samples into the performance evaluation. Additionally, exploring the reduction of IP constraints could further reduce computational complexity, providing valuable insights for future developments.
>
>
> > With FB one could use any arbitrary vector of random predictions (not necessarily from a classifier) and learn when to flip them or not. If so, why would one need a classifier in the first place?
>
> We agree with the reviewer from the theoretical point of view. Since our algorithm imposes no specific requirements on the baseline classifier, it can be any model within the hypothesis space and can have any random weights. Thus, in IP-Main, "flipping the labels of the baseline classifier" and "assigning labels directly in IP" are two ways to do the same thing, without a theoretical difference.
>
> However, from a practical perspective, our goal is to propose a post hoc method. It’s not that we "need" a classifier in the first space; rather, we primarily think of our work operating in a setting where a pretrained classifier already exists, and we aim to modify it to satisfy fairness and non-harmfulness in a post hoc manner. It's worth mentioning that, in real-world applications, baseline classifiers are often already with good accuracy performance. Thus, flipping the label of the existing classifier is more interpretable, generates more controlled predictions, and valuable insights may be gained by comparing the flipped results with those of the baseline classifier.
>
>
> > In the optimization problem in section 3.1 the abstention rate and the no harm constraints are not defined for any $z \in \mathcal{Z}$, whereas in the IP-Main these constraints are defined for each $z \in \mathcal{Z}$. If IP-Main is a way to practically solve the optimization problem in section 3.1, the definitions should be consistent. If there is a reason why these definitions should be different, this reason should be made clear.
>
> Constraints in Section 3.1 are defined for all $z \in \mathcal{Z}$, consistent with IP-Main. We have revised the representation in 3.1 for clarity.
>
> > Typos/Misc:
>
> We thank the reviewer for pointing them out and we make the change accordingly.

---

> > ### Comment · Reviewer_GsZB · 2023-11-22
> >
> > I would like to thank the authors for taking the time to address my concerns and to revise their work accordingly.

---

> > > ### Author Response · Authors · 2023-11-23
> > >
> > > Thank the reviewer for their response! We have also included additional experiments with more runs for Figure 3, along with the time cost and surrogate model training evaluation in Appendices E.7 and E.9.

---

### Official Review · Reviewer_JZKB · 2023-11-11

**Soundness:** 3 good
**Presentation:** 3 good
**Contribution:** 2 fair
**Rating:** 6
**Confidence:** 2

**Summary:**

The paper is considering the problem of enhancing fairness guarantees in model outputs. The specific problem considered is focusing on classification with abstention while increasing group fairness and maintaining model performance.
The authors argue that the previous approaches on fair classification or abstention don’t incorporate control over accuracy. They propose a 2-stage procedure to overcome this: (1) Integer Programming stage to generate abstention and flipping outcomes for each data point while maintaining accuracy, (2) Training a surrogate model against outputs of stage 1. They test their approach against two baselines on three real-world fairness datasets.

**Strengths:**

1. Paper is well structured and easy to read

2. The problem’s scope and methodology is well defined

3. The proposed method seems motivated; they seem to include the “no-harm” constraint along with giving feasibility conditions for disparity thresholds

4. The method is performant in the tasks considered

**Weaknesses:**

The reviewer is not convinced on the feasibility of the IP and the ability of surrogate to learn the patterns in AB or FB. Not a weakness as such, but would like to see a discussion from the authors.

**Questions:**

Could the authors show  the comparative performance on multi-group scenario in case of the Law and Compas datasets with the other baselines?

---

> ### Author Response · Authors · 2023-11-16
>
> We thank the reviewer for their positive feedback and constructive comments.
>
> > The reviewer is not convinced on the feasibility of the IP and the ability of surrogate to learn the patterns in AB or FB. Not a weakness as such, but would like to see a discussion from the authors.
>
> **Feasibility of IP.** Theoretically, we demonstrate that IP is not always feasible and investigate under what condition ($\varepsilon$, $\delta$, $\eta$) will it be feasible, under each fairness notion. For example, Theorem 4.1 establishes the minimum value of $\delta$ that is allowed, subject to upper bounds on disparity and a relaxation parameter for the error rate, under Demographic Parity. This highlights the importance of abstention by the more qualified group (higher qualification rate) for achieving a fair model without compromising accuracy, while the less qualified group need not abstain.
>
> We also run additional experiments on the feasibility region and verify the correctness of such linearity we proved. The figure can be found in Appendix E.8 in the updated version of our paper.
>
> **Ability of AB and FB to learn patterns.** The ability of surrogate models to learn patterns is empirically illustrated in Section 5, Table 2, by showing high accuracy. For example, when $\delta=0.2$ for Adult, AB achieves 92.20%, 89.93%, 88.48% accuracy under DP, EO, EOd, respectively, meanwhile FB achieves 97.79%, 95.33%, 95.86%. In addition, the significant improvement of FAN compared to baselines (Figure 3, 9, 11) also shows the effectiveness of both IP and surrogate model training. On a high level, in solving IP, we find that our solution encourages abstention and flipping on data samples that suffer from low confidence predictions and high uncertainties, thus the pattern can be captured by a sufficient surrogate model in Stage II.
>
>
> > Could the authors show the comparative performance on multi-group scenarios in case of the Law and Compas datasets with the other baselines?
>
> We’re running additional experiments and will revisit this question later before the end of the discussion phase.

---

> > ### Author Response · Authors · 2023-11-23
> >
> > We thank the reviewer again for their comments. We have conducted new experiments on multi-group scenarios and have attached the results in Appendix E.5 for the reviewer's reference.

---

### Meta-Review · Area_Chair_Ki74 · 2023-12-06

**Metareview:**

This paper tackles the problem of selective binary classification under fairness, abstention and harm constraints. All the reviewers are positive about the submission and the rebuttal convincingly addressed the comments/suggestions by the reviewers.

**Justification For Why Not Higher Score:**

Most of the reviewers are only mildly positive about the paper.

**Justification For Why Not Lower Score:**

None of the reviewers is negative about the paper and none of the points for improvement identified by the reviewers are major.

---

### Decision · Program_Chairs · 2024-01-16

Accept (poster)